# Learning Multiple Initial Solutions to Optimization Problems

## Abstract

Sequentially solving similar optimization problems under strict runtime constraints is essential for many applications, such as robot control, autonomous driving, and portfolio management. The performance of local optimization methods in these settings is sensitive to the initial solution: poor initialization can lead to slow convergence or suboptimal solutions. To address this challenge, we propose learning to predict *multiple* diverse initial solutions given parameters that define the problem instance. We introduce two strategies for utilizing multiple initial solutions: (i) a single-optimizer approach, where the most promising initial solution is chosen using a selection function, and (ii) a multiple-optimizers approach, where several optimizers, potentially run in parallel, are each initialized with a different solution, with the best solution chosen afterward. We validate our method on three optimal control benchmark tasks: cart-pole, reacher, and autonomous driving, using different optimizers: DDP, MPPI, and iLQR. We find significant and consistent improvement with our method across all evaluation settings and demonstrate that it efficiently scales with the number of initial solutions required.

## 1 Introduction

Many applications, ranging from trajectory optimization in robotics and autonomous driving to portfolio management in finance, require solving similar optimization problems sequentially under tight runtime constraints (Paden et al., 2016; Ye et al., 2020; Mugel et al., 2022). The performance of local optimizers in these contexts is often highly sensitive to the initial solution provided, where poor initialization can result in suboptimal solutions or failure to converge within the allowed time (Michalska & Mayne, 1993; Scokaert et al., 1999). The ability to consistently generate high-quality initial solutions is, therefore, essential for ensuring both performance and safety guarantees.

Conventional methods for selecting these initial solutions typically rely on heuristics or warm-starting, where the solution from a previously solved, related problem instance is reused. More recently, learning-based solutions have also been proposed, where neural networks are used to predict an initial solution. However, in more challenging cases, where the optimization landscape is highly non-convex or when consecutive problem instances rapidly change, predicting a single good initial solution is inherently difficult.

To this end, we propose *Learning Multiple Initial Solutions (MISO)* (Figure 1), in which we train a neural network to predict *multiple* initial solutions. Our approach facilitates two key settings: (i) a single-optimizer method, where a selection function leverages prior knowledge of the problem instance to identify the most promising initial solution, which is then supplied to the optimizer; and (ii) a multiple-optimizers method, where multiple initial solutions are generated jointly to support the execution of several optimizers, potentially running in parallel, with the best solution chosen afterward.

More specifically, our neural network receives a parameter vector that characterizes the problem instance and outputs $K$ candidate initial solutions. The network is trained on a dataset of problem instances paired with (near-)optimal solutions and is evaluated on previously unseen instances. Crucially, the network is designed not only to predict *good* initial solutions—those close to the optimal—but also to ensure that these solutions are sufficiently diverse, potentially spanning all underlying modes of the problem in hand. To actively encourage this multimodality, we implement training strategies such as a winner-takes-all loss that penalizes only the candidate with lowest

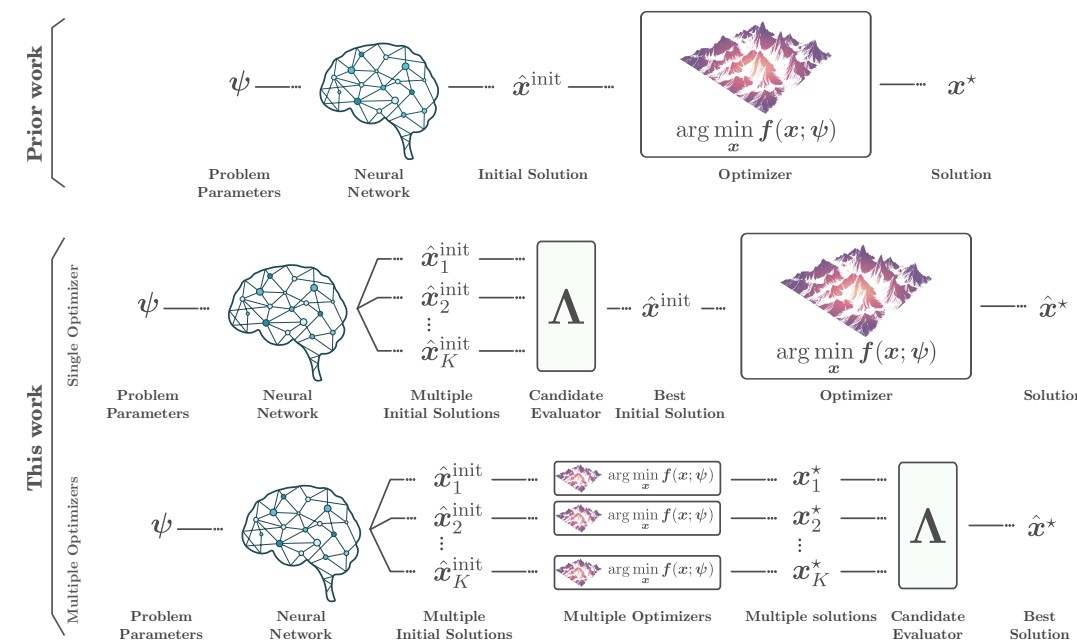

Figure 1: As opposed to previous works that predict a single initial solution, MISO trains a single neural network to predict *multiple* initial solutions. We use them to either initialize a single optimizer or jointly initialize multiple optimizers.

loss, a dispersion-based loss term to promote dispersion among solutions, and a combination of both.

We evaluate MISO across three distinct local optimization algorithms applied to separate robot control tasks: First-order Box Differential Dynamic Programming (DDP), which utilizes first-order linearization for the cart-pole swing-up task; Model Predictive Path Integral (MPPI) control, a sampling-based method, for the reacher task; and the Iterative Linear Quadratic Regulator (iLQR), a trajectory optimization algorithm, for an autonomous driving task. Our results show that MISO significantly outperforms existing initialization methods that rely on heuristics, learn to predict a single initial solution or use ensembles of independently learned models.

In summary, our key contributions are as follows:

1. We present a novel framework for predicting *multiple* initial solutions for optimizers.

2. We introduce two distinct strategies for utilizing the predicted initial solutions: (i) *single-optimizer*, where the most promising solution is chosen based on a selection function, and (ii) *multiple-optimizers*, where multiple optimizers are initialized, potentially in parallel, with the best solution chosen afterward.

3. We design and implement specific training objectives to prevent mode collapse and ensure that the predicted solutions remain multimodal.

4. We apply our framework to three distinct sequential optimization tasks and perform extensive evaluation.

## 2 RELATED WORK

**Learning for optimization.** Advancements in machine learning have introduced numerous learning-based approaches to optimization problems (Sun et al., 2019). Early work by Gregor & LeCun (2010) replaced components of classical convex optimization algorithms with neural networks. More recent works aim to replace optimization methods entirely with end-to-end neural networks (OpenAI et al., 2020; Mirowski et al., 2017) or generate new optimization algorithms (Chen et al., 2022b) for specific classes of problems. Other works enhance optimization-based control

algorithms (Sacks & Boots, 2022), learn constraints (Fajemisin et al., 2024), or learn objective functions and system dynamics (Lenz et al., 2015; Wahlström et al., 2015; Tamar et al., 2017; Hafner et al., 2019; Nagabandi et al., 2018; Xiao et al., 2022).

**Learning initial solutions.** Previous studies have proposed heuristic approaches to generate initial solutions for optimizers (Johnson et al., 2015; Marcucci & Tedrake, 2020). More recently, learning-based methods for initializing optimizers have gained attention in various fields, aiming to enhance both computational efficiency and resulting solutions quality. In mixed-integer programming, neural networks have enhanced solver performance by predicting variable assignments (Nair et al., 2020), branching decisions (Sonnerat et al., 2021), and integer variables (Bertsimas & Stellato, 2021). Baker (2019) employed Random Forests to predict solutions for AC optimal power flow problems. Kang et al. (2024) utilized nearest neighbor search to warm-start tight convex relaxations in nonconvex trajectory optimization problems. In robot control, neural networks were used to predict initializations for trajectory optimizers or Model Predictive Control (MPC) (Chen et al., 2022a; Wang & Ba, 2019; Lembono et al., 2020). An exciting line of recent work developed *differentiable* optimization algorithms, which allow jointly learning objectives, constraints, and initializations by backpropagating through the optimization process (Amos et al., 2018; East et al., 2019; Karkus et al., 2022; Sambharya et al., 2023). In contrast, we learn multiple initializations instead of one, and we do so without strong assumptions about the task or the optimizer. Notably, Bouzidi et al. (2023) used multiple initializations by repurposing a motion prediction model and Bézier curve fitting for a downstream MPC; however, this approach is specifically tailored for autonomous driving, incorporating a dedicated motion prediction module.

**Parallel optimizers.** Leveraging parallelism has a long history in optimization research (Betts & Huffman, 1991). With recent advances in parallel computing hardware, such as GPUs, methods that execute multiple optimizers in parallel have also emerged. For example, Sundaralingam et al. (2023) introduced cuRobo, a GPU-accelerated method combining L-BFGS and particle-based optimization for robotic manipulators. Similarly, Huang et al. (2024) utilized massive parallel GPU computation for efficient inverse kinematics and trajectory optimization. de Groot et al. (2024) proposed a topology-driven method that plans for multiple evasive maneuvers in parallel. Barcelos et al. (2024) focused on initializing parallel optimizers through rough paths. However, these works have not utilized learning. Lembono et al. (2020) explored learning-based strategies for initializing trajectory optimizers based on a database of previous solutions and ensemble-learned models, particularly in manipulation and humanoid control tasks. In contrast, we propose a single neural network to generate multiple initializations, which, as shown in our experiments, significantly outperforms the ensemble-based approach.

## 3 INITIALIZING OPTIMIZERS

**Problem setup.** In the most general form, we need to solve instances of a parameterized optimization problem,

$$\boldsymbol{x}^{\star}(\boldsymbol{\psi}) = \arg\min_{\boldsymbol{x}} J(\boldsymbol{x}; \boldsymbol{\psi}) \quad \text{s.t.} \quad \boldsymbol{g}(\boldsymbol{x}; \boldsymbol{\psi}) \leq \boldsymbol{0}; \quad \boldsymbol{h}(\boldsymbol{x}; \boldsymbol{\psi}) = \boldsymbol{0}, \tag{1}$$

where $\boldsymbol{x} \in \mathbb{R}^n$ is the variable vector to be optimized, $J$ is the objective function, $\boldsymbol{g}$ and $\boldsymbol{h}$ are collections of inequality and equality constraints, and $\boldsymbol{\psi} \in \mathbb{R}^m$ is a parameter vector that defines the problem instance, e.g., parameters of the objective function and constraints that differ across problem instances. A local optimization algorithm, **Opt**, attempts to find an optimum of $J$, namely,

$$\hat{\boldsymbol{x}}^{\star} = \mathbf{Opt}(J, \boldsymbol{\psi}, t_{\lim}; \boldsymbol{x}^{\text{init}}),$$

where $\boldsymbol{x}^{\text{init}}$ is initial solution provided to the optimizer, and $t_{\lim}$ is the runtime limit.

**Heuristic methods.** A common choice of the initial solution, $\boldsymbol{x}_{\text{init}}$, is the solution to a previously solved similar problem instance, referred to as a *warm-start*. For example, in optimal control the warm-start is typically the solution from the previous timestep, shifted and padded with zeros, $\boldsymbol{x}^{\text{w.s.}} := \{\{\boldsymbol{x}_{t+k}^{\text{cand}}\}_{k=1}^{H-2}, \boldsymbol{0}\}$ (Otta et al., 2015). This heuristic often works well in practice, however, it can struggle when large changes in the problem instance, $\boldsymbol{\psi}$, occur between consecutive time steps, leading to significant shifts in the optimal solution. For example, in autonomous driving, abrupt events like a traffic light switch or the sudden appearance of a pedestrian might drastically alter the reference trajectory or constraints. In such cases, the previous solution becomes a poor initialization, and the optimizer may fail to find a good solution within the allocated time frame.

## 4 LEARNING MULTIPLE INITIAL SOLUTIONS (👄 MISO)

The main idea of MISO is to train a single neural network to predict multiple initial solutions to an optimization problem, such that the initial solutions cover promising regions of the optimization landscape, eventually allowing a local optimizer to find a solution close to the global optimum. The key questions are then how to design a multi-output predictor; how to utilize multiple initial solutions in existing optimizers; and how to train the predictor to output a diverse set of initial solutions. In the following, we discuss our proposed solutions to these questions, illustrate the need for multimodality with a toy example, and discuss applications to optimal control.

### 4.1 MULTI-OUTPUT PREDICTOR

Our multi-output predictor is a standard Transformer model (Appendix A.3) that takes the problem instance, $\psi$, as input and outputs $K$ initial solutions for the optimization problem,

$$\{\hat{\boldsymbol{x}}_k^{\text{init}}\}_{k=1}^K = \boldsymbol{f}(\boldsymbol{\psi}; \boldsymbol{\theta}),$$

where $\boldsymbol{\theta}$ are the learned parameters of the network. We train the network on a dataset of problem instances and their corresponding (near-)optimal solutions, $\{(\boldsymbol{\psi}_i, \boldsymbol{x}_i^\star)\}_{i=1}^n$. Such dataset can be generated offline, for example, by running a slow yet globally optimal solver, or allowing the same local optimizer to run with longer time limits, potentially many times from different initial solutions.

### 4.2 OPTIMIZATION WITH MULTIPLE INITIAL SOLUTIONS

We propose two distinct settings to leverage multiple initial solutions: *single-optimizer* and *multiple-optimizers*. The resulting frameworks are illustrated in Fig. 1.

**Single optimizer.** In the single-optimizer setting we run a single instance of the optimizer with the most promising initial solution, $\hat{\boldsymbol{x}}^\star = \mathbf{Opt}(J, \boldsymbol{\psi}, t_{\text{limit}}; \hat{\boldsymbol{x}}^{\text{init}})$. We introduce a selection function, $\Lambda$, which, given a set of candidate solutions and the problem instance $\boldsymbol{\psi}$, returns the most promising candidate, $\hat{\boldsymbol{x}}^{\text{init}} = \Lambda(\{\hat{\boldsymbol{x}}_k^{\text{init}}\}_{k=1}^K, \boldsymbol{\psi})$. A reasonable choice for $\Lambda$ used in our experiments is selecting the candidate that minimizes the objective function the optimizer aims to minimize, i.e., $\Lambda := \arg\min_k J(\hat{\boldsymbol{x}}_k^{\text{init}}; \boldsymbol{\psi})$. Other possibilities include risk measures, metrics based on performance stability, robustness, exploration, or domain-specific metrics that align with the objectives of the overall task.

**Multiple optimizers.** In the multiple-optimizers setting, we assume multiple instances of the optimizer can be executed in parallel. We then initialize each optimizer with a different initial solution, $\boldsymbol{x}_k^\star = \mathbf{Opt}_k(J, \boldsymbol{\psi}, t_{\text{limit}}; \hat{\boldsymbol{x}}_k^{\text{init}}), \quad k \in \{1, \ldots, K\}$. To select a single solution from the outputs of the optimizers, we can use the same selection function $\Lambda$, as in the previous case, e.g., the solution that minimizes the objective function.

Our framework can be trivially generalized to allow a different number of optimizers and initial solution predictions, as well as using a heterogeneous set of optimization methods. Further, to maintain performance guarantees, one may include the default, e.g., warm-start, solution as one of the considered initial solutions, which ensures that even with poor predictions, the final solution quality does not degrade.

### 4.3 TRAINING STRATEGIES

The ultimate goal is to predict multiple initial solutions so that the downstream optimizer can find a solution close to the global optima, i.e., $J(\hat{\boldsymbol{x}}^\star; \boldsymbol{\psi}) \approx J(\boldsymbol{x}^\star; \boldsymbol{\psi})$. Training a neural network directly for this objective is not feasible in general. Instead, we propose proxy training objectives that combine two terms: a regression term that encourages outputs to be close to the global optimum, e.g., $\mathcal{L}_{\text{reg}}(\hat{\boldsymbol{x}}_k^{\text{init}}, \boldsymbol{x}^\star) = \|\hat{\boldsymbol{x}}_{\text{init}} - \boldsymbol{x}^\star\|$, where $\|\cdot\|$ is a distance metric; along with a *diversity* term that promotes outputs being different from each other, thereby covering various regions of the solution space. An illustrative example is in Sect. 4.4. In the following, we present three simple training strategies promoting diversity and preventing mode collapse. We discuss alternative formulations, with probabilistic modeling and reinforcement learning, in Sect. 7.

**Pairwise distance loss.** A simple method to encourage the model's outputs to differ from each other is to penalize the pairwise distance between all outputs. The overall loss combines this dispersion-promoting term with the regression loss,

$$\mathcal{L}_{\text{MISO-PD}} = \frac{1}{K}\sum_{k=1}^{K}\mathcal{L}_{\text{reg}}(\hat{\boldsymbol{x}}_k^{\text{init}}, \boldsymbol{x}^\star) + \alpha_K \frac{1}{K}\sum_{k=1}^{K}\mathcal{L}_{\text{PD},k}(\hat{\boldsymbol{x}}_k^{\text{init}}),$$

$$\mathcal{L}_{\text{PD},k} = \frac{1}{K-1}\sum_{\substack{k'=1 \\ k'\neq k}}^{K}\|\hat{\boldsymbol{x}}_k^{\text{init}} - \hat{\boldsymbol{x}}_{k'}^{\text{init}}\|,$$

where $\alpha_K$ is a hyperparameter that balances the trade-off between accuracy and dispersion.

**Winner-takes-all loss.** A more interesting way to encourage multimodality is to select the best-predicted output at training time and only minimize the regression loss for this specific prediction,

$$\mathcal{L}_{\text{MISO-WTA}} = \min_{k}\{\mathcal{L}_{\text{reg}}(\hat{\boldsymbol{x}}_k^{\text{init}}, \boldsymbol{x}^\star)\}.$$

Intuitively, the model only needs one of its outputs to be close to the ground truth, while the other predictions are not penalized for deviating, potentially aligning with different regions of the underlying distribution. Similar losses have been used, e.g., in multiple-choice learning (Guzman-Rivera et al., 2012). One advantage of this approach is that it is hyperparameter-free.

**Mixture loss.** Lastly, we consider a combination of the previous two approaches to potentially enhance performance, as it provides some measure of dispersion we can tune,

$$\mathcal{L}_{\text{MISO-MIX}} = \min_{k}\left\{\mathcal{L}_{\text{reg}}(\hat{\boldsymbol{x}}_k^{\text{init}}, \boldsymbol{x}^\star) + \alpha_K \Phi\left(\mathcal{L}_{\text{PD},k}(\hat{\boldsymbol{x}}_k^{\text{init}})\right)\right\},$$

here, $\Phi$ is an upper-bounded function, such as $\min$ or $\tanh$, designed to limit the contribution of the pairwise distance term.

Beyond the losses above, MISO could be integrated with other training paradigms, such as reinforcement learning or probabilistic modeling. We discuss these options in Sect. 7 but differ investigation to future work.

### 4.4 ILLUSTRATIVE EXAMPLE

To illustrate the advantage of using a single model with multimodal outputs compared to regression models or ensembles of regressors, we examine a straightforward one-dimensional optimization problem aimed at minimizing the cost function $c(x)$ shown in Fig. 2 (top). The function features two global minima, denoted as **A** and **C**, with a local minimum located between them at **B**.

Applying our learning framework to this simple problem, the dataset of optimal solutions includes instances of **A** and **C**. A single-output regression model has no means to distinguish the two modes and inevitably learns to predict the mean of examples in the dataset, somewhere near **B**. Consequently, the local optimizer is likely to converge to the suboptimal local minimum at **B**. Constructing an ensemble of such models to generate multiple initial solutions does not mitigate this issue, as each ensemble member tends to be biased toward the mean of the two modes near **B**. We implemented the optimization problem and showed the predictions for different training strategies in Fig. 2 (bottom). Details are in Appendix A.5. Indeed, an ensemble of single-output predictors fails to predict a global optimum, while our multi-output predictor succeeds with winner-takes-all and mixture losses.

While the problem considered here is purposefully simplistic, the existence of local minima is the key challenge in most optimization problems.

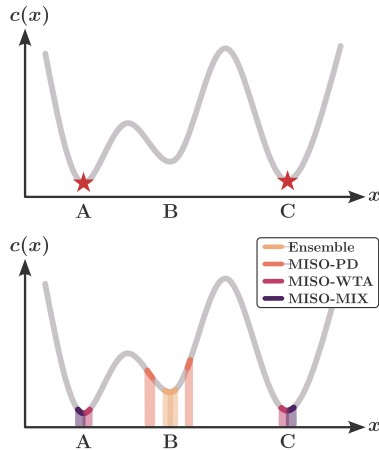

Figure 2: **Top**: The one-dimensional cost function $c(x)$ with global minima at **A** and **C** and a local minimum at **B**. **Bottom**: Predicted initial solution for different methods, demonstrating why explicitly promoting multimodality is important.

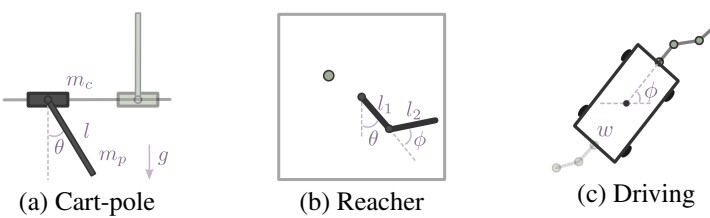

(a) Cart-pole  (b) Reacher  (c) Driving

Figure 3: Optimal control tasks used in our experiments.

### 4.5 APPLICATION TO OPTIMAL CONTROL

MISO is applicable to a broad class of sequential optimization problems; however, for the sake of evaluation, we focus on optimal control problems. Optimal control has a wide range of applications, e.g., in robotics, autonomous driving, and many other domains with strict runtime requirements, and due to the complexity induced by constraints and non-convex costs, local optimization algorithms are highly sensitive to the initial solution.

In optimal control the optimization variable $x$ represents a trajectory defined as a sequence of states and control inputs over discrete time steps: $\tau = \{s_t, u_t\}_{t=1:H}$. Here, $s_t \in \mathcal{S}$ and $u_t \in \mathcal{U}$ denote the state and control input at time step $t \in \mathbb{Z}^+$, and $H \in \mathbb{Z}^+$ is the optimization horizon. The constraints involve adhering to the system dynamics $f_d(s_{t+1}, s_t, u_t) = 0$, starting from an initial state $s_0 = s_{\text{curr}}$, where $s_{\text{curr}}$ represents the system's current state. The problem instance parameters $\psi$ encompass the initial state $s_0$, and other domain-specific variables that parameterize the objective function or constraints, such as target states, reference trajectories, obstacle positions, friction coefficients, temperature, etc.

A specific property of optimal control problems is that the relationship between optimization variables, states $s_t$ and controls $u_t$, are defined by the dynamics constraint $f_d$; and the initial state $s_0$ is given. Therefore, a sequence of controls uniquely defines an (initial) solution. We can leverage this property by learning to predict only a sequence of controls instead of the full optimization variable of state-control sequences. Further, one can define the training loss over either control, state, or state-control sequences and backpropagate gradients through the dynamics constraint as long as it is differentiable. In our experiments, we use state-control loss by default as we found it to improve both our and baseline learning methods. In Appendix A.7, we show that our conclusions hold with control-only loss as well.

## 5 EXPERIMENTAL SETUP

**Tasks.** We evaluated our method on the three robot control benchmark tasks shown in Fig. 3, each employing a distinct local optimization algorithm. **Cart-pole.** This task involves balancing a pole upright while moving a cart toward a randomly selected target position (Barto et al., 1983), using a first-order box Differential Dynamic Programming (DDP) optimizer (Amos et al., 2018). **Reacher.** In this task, a two-link planar robotic arm needs to reach a target placed at a random positon (Tassa et al., 2018), using a Model Predictive Path Integral (MPPI) optimizer (Williams et al., 2015). **Autonomous Driving.** Based on the nuPlan benchmark (Caesar et al., 2021), this task focuses on trajectory tracking in complex urban environments by following a reference trajectory generated by a Predictive Driver Model (PDM) planner (Dauner et al., 2023), using the Iterative Linear Quadratic Regulator (iLQR) optimizer (Li & Todorov, 2004). Further details are in Appendix A.1 and Appendix A.2.

**Baselines.** We compare MISO to a range of alternative methods to provide single or multiple initial solutions. For a single initial solution, we considered: **Warm-start**, the default method that uses the optimizer output from the last problem instance; **Regression**, a single-output regression model (the $K = 1$ version of MISO); **Oracle Proxy**, optimization with unlimited runtime, which we also used to generate our training data. For methods that generate multiple initial solutions, we considered: **Warm-start with perturbations**, which extends the warm-start approach by adding Gaussian noise to the optimizer output from the last problem instance; **Regression with perturbations**, where Gaussian noise is introduced to the predictions of the single-output regression model; **Multi-output**

Table 1: Results for the single optimizer setting. The mean cost of solutions found by the single optimizer using different initial solutions across tasks and evaluation settings.

| Method | $K$ | One-Off Optimization | | | Sequential Optimization | | |
|---|---|---|---|---|---|---|---|
| | | Reacher | Cart-pole | Driving | Reacher | Cart-pole | Driving |
| **Single Optimizer** | | | | | | | |
| Warm-start | 1 | 13.48 ±0.88 | 11.69 ±0.84 | 283.86 ±37.91 | 13.48 ±0.88 | 11.69 ±0.84 | 283.86 ±37.91 |
| Regression | 1 | 13.40 ±0.88 | 11.19 ±0.80 | 74.23 ±7.69 | 19.56 ±0.52 | 6.18 ±0.47 | 70.62 ±7.38 |
| Warm-start w. perturb | 32 | 13.46 ±0.88 | 11.64 ±0.83 | 145.01 ±23.01 | 14.71 ±0.93 | 16.29 ±0.44 | 164.75 ±22.84 |
| Regression w. perturb | 32 | 13.38 ±0.88 | 11.16 ±0.80 | 67.69 ±8.01 | 15.28 ±0.58 | 5.74 ±0.47 | 66.75 ±6.56 |
| Multi-output regression | 32 | 13.41 ±0.88 | 11.21 ±0.80 | 70.25 ±8.75 | 18.49 ±0.55 | 6.62 ±0.45 | 78.74 ±8.99 |
| Ensemble | 32 | 13.39 ±0.88 | 10.94 ±0.79 | 47.22 ±4.71 | 8.40 ±0.40 | 3.55 ±0.34 | 52.59 ±4.81 |
| MISO pairwise dist. | 32 | 13.41 ±0.88 | 11.22 ±0.80 | 66.06 ±7.48 | 19.20 ±0.49 | 6.07 ±0.45 | 71.90 ±7.90 |
| MISO winner-takes-all | 32 | 13.36 ±0.88 | **10.48 ±0.77** | 30.17 ±2.24 | 2.72 ±0.21 | 0.83 ±0.06 | **30.75 ±2.15** |
| MISO mix | 32 | **12.74 ±0.86** | **10.48 ±0.77** | 33.95 ±2.39 | **2.44 ±0.20** | **0.79 ±0.04** | 33.38 ±2.21 |
| Oracle Proxy | 1 | 13.43 ±0.88 | 11.01 ±0.80 | 41.94 ±4.31 | 6.88 ±0.58 | 4.54 ±0.71 | 26.52 ±2.00 |

Table 2: Results for the multiple optimizers setting. Mean cost of solutions found by multiple optimizers using different initial solutions across tasks and evaluation settings.

| Method | $K$ | One-Off Optimization | | | Sequential Optimization | | |
|---|---|---|---|---|---|---|---|
| | | Reacher | Cart-pole | Driving | Reacher | Cart-pole | Driving |
| **Multiple Optimizers** | | | | | | | |
| Warm-start w. perturb | 32 | 13.41 ±0.88 | 10.93 ±0.79 | 155.53 ±24.33 | 5.89 ±0.50 | 6.68 ±0.59 | 162.13 ±34.10 |
| Regression w. perturb | 32 | 13.34 ±0.88 | 11.12 ±0.80 | 64.88 ±6.84 | 3.53 ±0.27 | 5.36 ±0.48 | 62.07 ±6.39 |
| Multi-output regression | 32 | 13.34 ±0.88 | 11.21 ±0.80 | 70.29 ±9.13 | 3.31 ±0.26 | 6.38 ±0.43 | 70.71 ±8.18 |
| Ensemble | 32 | 13.34 ±0.88 | 10.65 ±0.78 | 45.44 ±4.64 | 3.08 ±0.23 | 2.21 ±0.20 | 49.08 ±5.29 |
| MISO pairwise dist. | 32 | 13.34 ±0.88 | 11.22 ±0.80 | 67.62 ±7.58 | 3.42 ±0.27 | 6.09 ±0.47 | 71.33 ±8.13 |
| MISO winner-takes-all | 32 | 13.34 ±0.88 | **10.29 ±0.76** | 30.87 ±2.30 | 2.21 ±0.16 | 0.76 ±0.05 | **30.48 ±2.07** |
| MISO mix | 32 | **12.72 ±0.86** | **10.29 ±0.76** | 33.52 ±2.35 | **1.56 ±0.14** | **0.63 ±0.02** | 34.85 ±2.64 |

**regression**, a naive multi-output regression model without a diversity-promoting objective; and **Ensemble**, which trains multiple single-output neural networks with different random initializations. Finally, we assessed variants of our proposed method with the different training losses discussed in Sect. 4.3: **pairwise distance**, **winner-takes-all**, and **mix**.

**Evaluation settings.** We employ two evaluation modes. (i) **One-off**, where the optimization task is treated as an isolated problem with the objective of finding the minimum of a given function. This mode serves as the default configuration for training neural networks, where data is replayed to the model, and the optimizer's solution is recorded but not executed. Methods are assessed by the mean cost of the optimizer's output over problem instances. (ii) **Sequential**, which involves solving a series of related optimization problems, executing each proposed solution, and starting the subsequent optimization from the resulting state. This setting simulates real-world conditions where the optimizer continuously interacts with a dynamic environment in a closed loop. We evaluate performance by taking the mean cost over problems in a sequence, and then the mean over sequences.

To account for the additional time required to predict initial solutions, we assumed that all models perform inference in under 0.85ms, which was the case for all methods on both CPU and GPU, except for the ensemble (see Appendix A.6). In the autonomous driving task, we then reduced the runtime allocated to the optimizer accordingly.

**Implementation details.** To generate the training data, we first create a set of problem instances by sequentially executing the optimizer initialized with the default warm-start strategy. The problem instances are then fed again to an "oracle" version of the optimizer with a significantly increased runtime limit, and the resulting solutions are recorded. After training, evaluation is done on a separate unseen set of problem instances. All experiments are conducted on an Intel Core i9-13900KF CPU and an NVIDIA RTX 4090 GPU. Further implementation details, including hyperparameters and training procedures, are in Appendix A.4 and Appendix A.3.

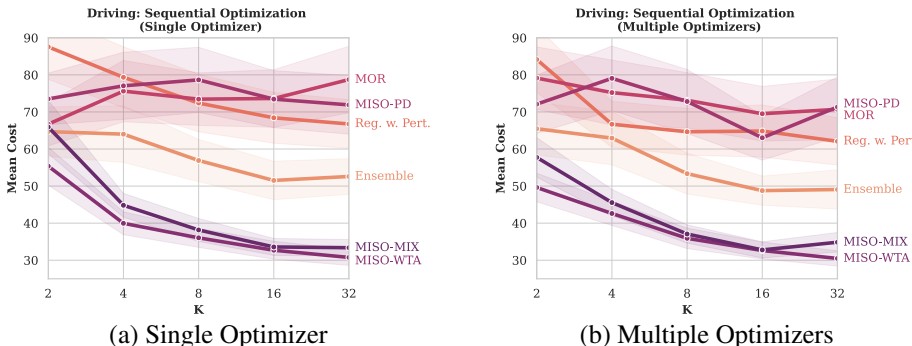

(a) Single Optimizer              (b) Multiple Optimizers

Figure 4: Mean Cost of the autonomous driving environment with varying values of $K$. The shaded regions around each curve indicate the standard error of the mean.

## 6 RESULTS

Our main results for optimization with different initial solutions are reported in Table 1 and Table 2 for single optimizer and multiple optimizers settings, respectively. Figure 4 shows the effect of the number of predicted initial solutions. Figure 5 provides qualitative results. More detailed results, including inference times, are in the Appendix.

**Single optimizer**. In the single-optimizer setting, Table 1, we first observe that even one learned initialization outperforms heuristic solutions (regression vs. warm-start), in almost all settings, and in particular in the most challenging autonomous driving task. We then examine the impact of generating multiple initial solutions. Perturbations-based methods show some improvement over their single-initialization counterparts in most cases, and ensembles of independently learned models perform consistently better than single models. Finally, our proposed multi-output methods demonstrate substantial improvements over all baselines because they can learn to predict diverse multimodal initial solutions. Specifically, MISO winner-takes-all or MISO mix achieve the lowest mean costs across all tasks. Considering the pairwise distance term alone proves insufficient to ensure adequate diversity, whereas incorporating it with MISO winner-takes-all often boosts performance, yet, its effectiveness varies, which underscores the challenge of selecting optimal hyperparameters. As expected, improvements are consistently larger in the more important sequential optimization setting, where errors over time compound.

**Multiple optimizers**. When considering the multiple-optimizers setting, we observe the same trend. Learning-based methods outperform heuristic ones, and multi-output approaches yield further enhancements. As expected, the use of multiple optimizers leads to consistently better results compared to the single-optimizer setting due to increased exploration of the solution space.

**Scaling with the number of initial solutions.** Figure 4 shows that our method scales effectively and consistently with the number of predicted initial solutions $K$, and outperforms other approaches across varying values of $K$. Importantly, as $K$ increases, the inference time for ensemble approaches grows, whereas MISO remains almost constant (see Appendix A.6). We further evaluate mode diversity in Appendix A.8, and find that, in line with our conclusions, all MISO outputs remain useful even when $K$ increases.

**Qualitative results.** Figure 5 (left) depicts the optimizer's output trajectories with different initial solutions for the autonomous driving task. In this scenario, the high-level planner abruptly alters the reference path, which could happen, e.g., because of a newly detected pedestrian. The change in reference path makes the previous solution (warm-start) a poor initialization, and the optimizer converges to a local minimum that minimizes control effort but is far from the desired path. Regression and model ensemble also fail to predict a good initial solution. In contrast, MISO winner-takes-all adapts to this sudden reference change and closely follows the reference path. Figure 5 (right) depicts MISO's initial solutions for the cart-pole task. The different outputs capture different modes of the solution space (moving upright, moving with left swing, moving with right swing), showing MISO's ability to generate diverse and multimodal solutions.

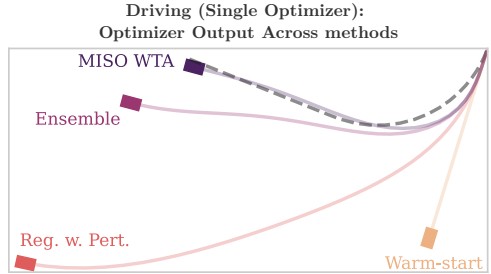
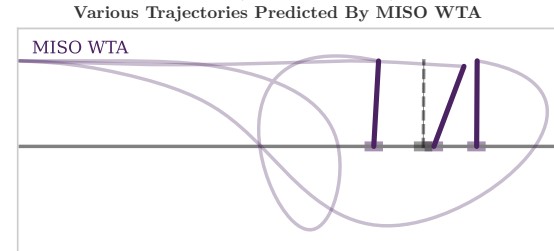

Figure 5: Left: Each method adaptation when the high-level planner abruptly modifies the reference path. Right: Multiple predicted trajectories of MISO winner-takes-all.

**Summary.** Overall, our methods significantly outperform the other baselines in both settings. The consistent superiority of the MISO mix and MISO winner-takes-all methods across different tasks and configurations underscores the advantages of using learning-based multi-output strategies for generating initial solutions. These findings demonstrate that promoting diversity among multiple initializations is crucial for improving optimization outcomes, especially when combined with multiple optimizers.

## 7 CONCLUSIONS AND FUTURE WORK

We introduced Learning Multiple Initial Solutions (MISO), a novel framework for learning multiple diverse initial solutions that significantly enhance the reliability and efficiency of local optimization algorithms across various settings. Extensive experiments in optimal control demonstrated that our method consistently outperforms baseline approaches and scales efficiently with the number of initializations.

**Limitations.** Our approach is not without limitations. First, to train a useful model, we rely on the coverage and quality of the training data, as the method does not directly interact with the optimizer or the underlying objective function. Second, the underlying assumption of our regression loss is that initial solutions closer to the global optimum increase the likelihood of successful optimization may not hold in complex optimization landscapes with intricate constraints. Third, in highly complex optimization problems where each solution constitutes a high-dimensional and intricate structure, accurately learning initial solution candidates can become exceedingly challenging, potentially diminishing the effectiveness of our approach.

**Future work.** There are several promising directions for future research. To address the aforementioned limitations, one may simply incorporate the optimization objective into the model training loss, thus creating a direct link to the final optimization goal. Alternatively, using reinforcement learning (RL) to train MISO is a particularly exciting opportunity. By framing the problem in an RL context, e.g., where the reward is the negative cost of the optimizer's final solution, models would be directly trained to maximize the probability of the optimizer finding the global optima and may learn to specialize to the specific optimizer. One challenge would be computational, as RL would require running the optimizer numerous times during training.

Other extensions of our approach include probabilistic modeling, e.g., Gaussian mixture models, variational autoencoders, or diffusion models; however, preventing mode collapse and promoting diversity would remain a challenge. Future work may explore alternative selection functions, such as risk measures or criteria based on stability, robustness, exploration, or other domain-specific metrics; as well as using a heterogeneous set of parallel optimizers. Finally, we are excited about various possible applications in optimal control and beyond, where sequences of similar optimization problems need to be solved, for example, localization and mapping in robotics, financial optimization, traffic routing optimization, or even training neural networks with different initial weights, e.g., for meta-learning, or scene representation learning with Neural Radiance Fields or 3D Gaussian splatting.

**Ethics statement.** Our work is concerned with a general class of optimization problems that do not raise particular ethical considerations.

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

## A APPENDIX

### A.1 DETAILED DESCRIPTIONS OF BASELINE OPTIMIZERS

This subsection provides detailed descriptions of the optimization algorithms used in our evaluations: First-order Box Differential Dynamic Programming (DDP), Model Predictive Path Integral (MPPI), and the Iterative Linear Quadratic Regulator (iLQR). These algorithms were selected due to their widespread use and effectiveness in solving optimal control problems across various domains.

**First-order Box Differential Dynamic Programming (DDP).** Building on the work of Tassa et al. (2014), Amos et al. (2018) introduced a simplified version of Box-DDP that utilizes first-order linearization instead of second-order derivatives. This approach, termed "first-order Box-DDP," reduces computational complexity while maintaining the ability to handle box constraints on both the state and control spaces.

**Model Predictive Path Integral (MPPI).** MPPI (Williams et al., 2015) is a sampling-based model predictive control algorithm that iteratively refines control inputs using stochastic sampling. Starting from the current state and a prior solution, it generates a set of randomly perturbed control sequences, simulates their trajectories, and evaluates them using a cost function. The control inputs are then updated based on a weighted average, favoring lower-cost trajectories. We use the implementation from `https://github.com/UM-ARM-Lab/pytorch_mppi`.

**Iterative Linear Quadratic Regulator (iLQR).** iLQR (Li & Todorov, 2004) is a trajectory optimization algorithm that refines control sequences iteratively by linearizing system dynamics and approximating the cost function quadratically around a nominal trajectory. It alternates between a forward pass, simulating the system trajectory, and a backward pass, computing optimal control updates. We use the implementation provided in the nuPlan simulator.

### A.2 DETAILED TASK DESCRIPTIONS AND HYPERPARAMETERS

This subsection provides detailed descriptions of the tasks used in our experiments: cart-pole, reacher, and autonomous driving. For each task, we outline the system dynamics, control inputs, and the specific hyperparameters employed in our evaluations.

**Cart-pole.** The cart-pole task (Barto et al., 1983) involves a cart-pole system tasked with swinging the pole upright while moving the cart to a randomly selected target position along the rail. The goal is to balance the pole vertically and simultaneously reach the target cart position. The system is characterized by the state vector $\boldsymbol{s}_t \in \mathbb{R}^4$, which includes the pole angle $\theta$, pole angular velocity $\dot{\theta}$, cart position $x$, and cart velocity $\dot{x}$. The control input is a single force applied to the cart, $\boldsymbol{u}_t \in \mathbb{R}$.

**Hyperparameters.** The mass of the cart is $m_c = 1.0\,\mathrm{kg}$, the mass of the pole is $m_p = 0.3\,\mathrm{kg}$, and the length of the pole is $l = 0.5\,\mathrm{m}$. Gravity is set to $g = -9.81\,\mathrm{m/s}^2$. Control inputs are bounded by $u_{\min} = -5.5\,\mathrm{N}$ and $u_{\max} = 5.5\,\mathrm{N}$, with a time step of $\Delta t = 0.1\,\mathrm{s}$ and $n_{\text{sub\_steps}} = 2$ physics sub-steps per control step. Each episode has a maximum length of $T_{\text{env}} = 50$ steps. Both optimizers use goal weights of $[0.1, 0.01, 1.0, 0.01]$ for the state variables and a control weight of $0.0001$. The prediction horizon is set to $H = 10$. For the online optimizer, we set `lqr_iter` = 2 and `max_linesearch_iter` = 1. In the oracle optimizer, we use `lqr_iter` = 10 and `max_linesearch_iter` = 3. The initial state $\boldsymbol{s}_0 \in \mathbb{R}^4$ is sampled as follows: $x_0 \sim \mathcal{U}(-2, 2)\,\mathrm{m}$, $\dot{x}_0 \sim \mathcal{U}(-1, 1)\,\mathrm{m/s}$, $\theta_0 \sim \mathcal{U}(-\frac{\pi}{2}, \frac{\pi}{2})\,\mathrm{rad}$, and $\dot{\theta}_0 \sim \mathcal{U}(-\frac{\pi}{4}, \frac{\pi}{4})\,\mathrm{rad/s}$. The goal state is defined by a target cart position $x_{\text{goal}} \sim \mathcal{U}(-2, 2)\,\mathrm{m}$, while the rest of the state variables (pole angle and velocities) are set to zero, ensuring the goal is to bring the pole upright and bring the system to rest.

**Reacher.** The Reacher task (Tassa et al., 2018) involves a two-link planar robot arm tasked with reaching a randomly positioned target. The goal is to move the end-effector to the target position in the plane. The system is characterized by the state vector $\boldsymbol{s}_t \in \mathbb{R}^4$, which includes the joint angles $\theta_1, \theta_2$ and angular velocities $\dot{\theta}_1, \dot{\theta}_2$. The control inputs, $\boldsymbol{u}_t \in \mathbb{R}^2$, are torques applied to each joint.

**Hyperparameters.** The simulation uses a time step of $\Delta t = 0.02\,\mathrm{s}$, with joint damping set to $0.01$ and motor gear ratios of $0.05$. The control inputs are constrained by $u_{\min} = [-1, -1]$ and $u_{\max} = [1, 1]$. The wrist joint has a limited range of $[-160°, 160°]$. Each episode is limited to $T_{\text{env}} = 250$ steps. Both optimizers are set with a control noise covariance $\sigma^2 = 1 \times 10^{-3}$, a

temperature parameter $\lambda = 1 \times 10^{-4}$, and a prediction horizon $H = 10$. The online optimizer uses `num_samples = 3`, while the oracle optimizer uses `num_samples = 50`. The target position is generated by sampling $\theta_{\text{target}} \sim \mathcal{U}(0, 2\pi)$ and $r_{\text{target}} \sim \mathcal{U}(0.05, 0.20)\,\text{m}$, then set as $\text{pos}_x = r_{\text{target}} \cos(\theta_{\text{target}})$ and $\text{pos}_y = r_{\text{target}} \sin(\theta_{\text{target}})$.

**Autonomous Driving.** The autonomous driving task, based on the nuPlan benchmark (Caesar et al., 2021), evaluates the performance of motion planning algorithms in complex urban environments. Our focus is on the control (tracking) layer, which is responsible for accurately following the planned trajectories. The task involves navigating a vehicle through a series of scenarios with varying traffic conditions, obstacles, and road layouts. The goal is to execute safe, efficient, and comfortable trajectories while adhering to traffic rules and avoiding collisions. The system is characterized by the state vector $s_t \in \mathbb{R}^5$, which includes the vehicle's position $(x, y)$, orientation $\phi$, velocity $v$, and steering angle $\delta$. The control inputs, $u_t \in \mathbb{R}^2$, are acceleration $a$ and steering angle rate $\dot{\delta}$. We use the state-of-the-art Predictive Driver Model (PDM) planner (Dauner et al., 2023) to generate the reference trajectories $r_t$.

**Hyperparameters.** The prediction horizon is set to $H = 40$ with a discretization time step of $\Delta t = 0.2\,\text{s}$. The cost function is weighted with state cost diagonal entries $[1.0, 1.0, 10.0, 0.0, 0.0]$ for the position, heading, velocity, and steering angle, respectively, and input cost diagonal entries $[1.0, 10.0]$ for acceleration and steering angle rate. The maximum acceleration is constrained to $3.0\,\text{m/s}^2$, the maximum steering angle is $60°$, and the maximum steering angle rate is $0.5\,\text{rad/s}$. A minimum velocity threshold for linearization is set at $0.01\,\text{m/s}$. The online optimizer is limited to a maximum solve time of `max_solve_time = 5\,ms`, while the oracle optimizer allows for `max_solve_time = 50\,ms`. For a fair comparison, we keep the total runtime limit fixed, including both initialization and optimization. The total runtime limit for warm-start and perturbation methods is $5\,\text{ms}$. For learning-based methods, the inference time for our neural networks is between $0.6\,\text{ms}$ and $0.7\,\text{ms}$ on a GPU, and $0.7\,\text{ms}$ to $0.8\,\text{ms}$ on a CPU. For simplicity, we allocate $0.85\,\text{ms}$ for model inference and run the optimizer for the remaining $4.15\,\text{ms}$. We have not performed any inference optimization for our models (e.g., TensorRT).

## A.3 NETWORK ARCHITECTURE AND TRAINING DETAILS

This section provides a brief overview of the network architecture, the data collection process, and the training procedures used in our experiments. We summarize the design of our base Transformer model, outline the methods used to generate and preprocess the training data, and detail the key training methodologies and hyperparameters employed.

**Network Architecture.** The base model is a standard Transformer architecture with absolute positional embedding, a linear decoder layer, and output scaling. The core Transformer architecture remains standard, with task-specific configurations. The input to the network is the concatenated sequence of the warm-start state trajectory error, $\tau_e^{\text{w.s.}}$, defined as the difference between the reference trajectory, $\psi = \tau_r$, (in the autonomous driving task) or the goal state, $\psi = x_g$ (in the cart-pole and reacher tasks), and the warm-start trajectory, $\tau_x^{\text{w.s.}}$. The warm-start control trajectory is $\tau_u^{\text{w.s.}} = \{\{u_{t+k}^{\text{cand}}\}_{k=1}^{H-2}, \mathbf{0}\}$. The network predicts $K$ control trajectories, $\{\hat{\tau}_{u,k}^{\text{init}}\}_{k=1}^K$, for the next optimization step. Each environment's configuration is described in Table 3.

Table 3: Configuration of the Transformer model for each environment.

| Parameter | Description | Reacher | Cart-pole | Driving |
|---|---|---|---|---|
| `n_layer` | Number of layers | 4 | 4 | 4 |
| `n_head` | Number of heads | 2 | 2 | 2 |
| `n_embd` | Embedding dimension | 64 | 64 | 64 |
| `dropout` | Dropout rate | 0.1 | 0.1 | 0.1 |
| `src_dim` | Input dimension | 8 | 5 | 7 |
| `src_len` | Sequence length | 10 | 9 | 40 |
| `out_dim` | Output dimension | 2 | 1 | 2 |

**Data Collection and Preprocessing.** In all experiments, the training data is generated by (1) unrolling an optimizer using a warm-start initialization policy and recording its inputs and outputs, and (2) replaying the same scenarios using an oracle optimizer—essentially the same optimizer with

enhanced capabilities, such as more optimization steps or additional sampled trajectories—and logging its inputs and outputs. The resulting mapping may be seen as a filter, refining (near-)optimal initial solutions into optimal ones. For each task, 500,000 instances were collected.

**Training.** Prior to training, all features were standardized to ensure consistent input scaling. We used the AdamW optimizer and applied gradient norm clipping. The models were trained using standard settings without any complex modifications. All hyperparameters are detailed in Table 4.

Table 4: Training hyperparameters for each environment.

| Parameter | Description | Reacher | Cart-pole | Driving |
|---|---|---|---|---|
| epochs | Epochs | 125 | 125 | 125 |
| batch_size | Batch size | 1024 | 1024 | 1024 |
| lr | Learning rate | 0.001 | 0.0003 | 0.0001 |
| weight_decay | Weight decay | 0.0001 | 0.0001 | 0.0001 |
| grad_norm_clip | Gradient norm clipping | 2.0 | 2.0 | 2.0 |
| control_loss_weight | Control loss weight | 100.0 | 1.0 | 5.0 |
| state_loss_weight | State loss weight | 0 | 0.01 | 0.005 |
| pairwise_loss_weight | Pairwise loss weight | 0.1 | 0.01 | 0.1 |

## A.4 DESCRIPTIONS OF BASELINE METHODS

This section introduces the baseline methods used in our experiments in more detail and discusses their implementation specifics. These baselines serve as reference points to evaluate our proposed methods' performance and understand the benefits and limitations of different initialization strategies in optimization algorithms.

**Warm-start** ($K = 1$). A common technique involves shifting the previous solution forward by one time step and padding it with zeros. Assuming the system does not exhibit rapid changes in this interval, the previous solution should retain local, feasible information.

**Oracle Proxy** ($K = 1$). This proxy serves two purposes: (1) estimating the gap between a real-time-constrained optimizer and an unrestricted one and (2) providing a proxy for a mapping worth learning. For each optimization algorithm, a suitable heuristic is defined. In DDP, the oracle is allowed more iterations to converge; in MPPI, it has a larger sample budget; and in iLQR, it is given more time to perform optimization iterations.

**Regression** ($K = 1$). This approach involves training a neural network to approximate the oracle's mapping. Unlike the oracle heuristic, which is impractical for real-time use, the trained neural network requires only a single forward pass.

**Warm-start with Perturbations** ($K > 1$). We utilize the warm-start technique as another baseline by duplicating the proposed initial solution $K$ times and adding Gaussian noise. While this introduces some form of dispersion, the resulting initial solutions are neither guaranteed to be feasible nor to ensure any level of optimality.

**Regression with Perturbations** ($K > 1$). Similar to the warm-start with perturbation, after predicting an initial solution using the neural network, we duplicate it $K$ times and add perturbations.

**Ensemble** ($K > 1$). An ensemble of $K$ separate neural networks leverages the idea that networks initialized with different weights during training will often produce different predictions. The main drawbacks of this approach are (1) the need to train $K$ neural networks and (2) the requirement to run $K$ forward passes, which may be impractical for real-time deployment.

**Multi-Output Regression** ($K > 1$). A naive approach to predicting $K$ initializations from a single network involves calculating the loss for each prediction and summing the losses. However, since there is no explicit multimodality objective, these models are prone to mode collapse.

**MISO Pairwise Distance** ($K > 1$). One way to mitigate mode collapse is to introduce an additional term in the loss function, such as the pairwise distance between predictions. While this approach requires weight tuning and selecting an appropriate norm, a significant challenge lies in understanding how effectively this term promotes multimodality in practice.

**MISO Winner-Takes-All** ($K > 1$). This approach updates only the best-performing mode based on the loss of each prediction. Although no explicit dispersion objective is included, multimodality is indirectly encouraged by maintaining multiple active modes while refining only the best one and not penalizing the others.

**MISO Mix** ($K > 1$). Lastly, we combine the pairwise distance term with the Winner-Takes-All approach. While this allows for greater refinement, it adds complexity due to additional hyperparameters and the need to apply further operations—such as clamping distances—to ensure the loss won't diverge.

## A.5 ILLUSTRATIVE EXAMPLE

This example provides a simplified scenario to illustrate the behavior of local optimization algorithms in a controlled, low-dimensional setting with non-convex and multimodal objective function. The system dynamics are defined by the linear equation $x_{t+1} = x_t + u_t$, where $x_t \in \mathbb{R}$ represents the state and $u_t \in [-1, 1]$ is the constrained control input at time step $t$. The initial state is $x_0 = 0$, and the optimization horizon is set to $H = 5$. The objective is to find the optimal control trajectory $\boldsymbol{\tau}_u^\star = \{u_t\}_{t=0}^{H-1}$ that minimizes the cumulative cost function $\boldsymbol{\tau}_u^\star = \arg\min_{\boldsymbol{\tau}_u} \sum_{k=0}^{H-1} c(x_{t+k})$, where the resulting state trajectory $\boldsymbol{\tau}_x = \{x_t\}_{t=0}^{H}$ is obtained by unrolling $\boldsymbol{\tau}_u$ from $x_0$. The cost function, $c(x) = (x^2 + 0.05)(x + 1.5)^2(x - 2)^2$, is non-convex and multimodal, featuring two global minima at $x_1^\star = -1.5$ and $x_2^\star = 2$.

Table 5: Comparison of different methods for predicting the optimal control trajectories

| Method | $x_{H+1}$ | $\hat{\tau}_u$ |
|---|---|---|
| Ensemble | 0.03 | -0.02, 0.03, 0.00, 0.01, 0.01 |
| | 0.02 | -0.01, 0.02, -0.01, 0.01, 0.01 |
| MISO pairwise dist. | -0.24 | -0.12, -0.05, -0.04, -0.02, 0.01 |
| | 0.24 | 0.08, 0.05, 0.04, 0.03, 0.03 |
| MISO winner-takes-all | **-1.49** | -0.88, -0.65, 0.00, 0.02, 0.02 |
| | **2.01** | 0.99, 0.94, 0.05, 0.01, 0.01 |
| MISO mix | -1.52 | -0.90, -0.68, 0.00, 0.03, 0.03 |
| | 2.05 | 0.99, 0.95, 0.07, 0.02, 0.02 |
| Optimal | -1.50 | -1.00, -0.50, 0.00, 0.00, 0.00 |
| | 2.00 | 1.00, 1.00, 0.00, 0.00, 0.00 |

The results from Table 5 provide a comparison of different methods for predicting optimal control trajectories. The Ensemble method, which combines multiple single-output predictions, yields the least accurate results, with final states close to zero. Due to the dispersion term, MISO pairwise distance is a bit further from zero but still far from either optimum. On the other hand, MISO winner-takes-all and MISO mix successfully predict both optimal sequences with high fidelity and thus are able to reach either global optimum.

Overall, the results suggest that methods specifically tailored for capturing multimodality, such as the winner-takes-all and mixed strategies, are more effective than their single-output regression counterparts, particularly in non-convex environments.

## A.6 INFERENCE TIME: ENSEMBLE VS. MULTI-OUTPUT MODELS

This experiment benchmarks the execution time of two model architectures: an ensemble of $K$ single-output models and a single multi-output model producing $K$ outputs. Both models are based on the Transformer architecture used in the autonomous driving environment.

The experiments were conducted on an Intel Core i9-13900KF CPU and an NVIDIA RTX 4090 GPU, measuring the mean inference time over 1000 runs across five random seeds. GPU operations were synchronized before timing to ensure accurate measurements. The results, shown in Fig. 6, display the mean inference time in milliseconds as $K$ increases for both the ensemble and multi-output models on CPU and GPU.

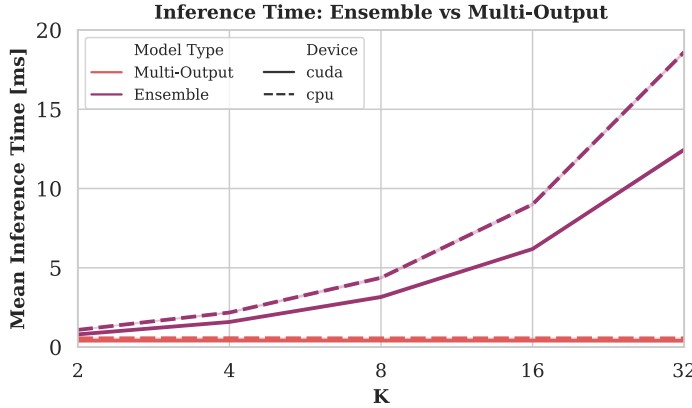

Figure 6: Mean inference time for varying values of $K$ for the ensemble and multi-output models on CPU and CUDA

The multi-output model exhibits minimal sensitivity to the increase in $K$ on both the CPU and GPU, indicating that this architecture scales efficiently, maintaining a low overhead even as the number of outputs grows. In contrast, the ensemble model's inference time increases significantly with $K$, suggesting that managing multiple models introduces overhead that scales poorly as $K$ grows.

In applications with strict runtime constraints, such as the autonomous driving environment, the ensemble approach becomes impractical as $K$ increases. Conversely, the multi-output model remains a viable option, even at larger values of $K$, making it the preferred choice for time-sensitive scenarios.

### A.7 STATE LOSS

An additional challenge in learning control policies is addressing *compounding errors*—small inaccuracies in the predicted control trajectory $\tau_u$ that, when unrolled, cause significant deviations in the state trajectory $\tau_x$. Even if most elements of $\tau_u$ are accurate, errors in the initial steps can cause the state $\tau_x$ to drift, leading to further divergence as the system evolves.

To mitigate compounding errors, we introduce a regression loss not only over the control trajectory $\tau_u$ but also over the resulting state trajectory $\tau_x$. In a supervised learning setting, this requires a model of the system dynamics, which can either be known or learned.

Let $\hat{\tau}_u = \{\hat{u}_t\}_{t=0}^{H-1}$ denote the predicted control trajectory, and $\tau_u^\star = \{u_t^\star\}_{t=0}^{H-1}$ denote the target control trajectory. Similarly, let $\hat{\tau}_x = \{\hat{x}_t\}_{t=1}^{H}$ be the predicted state trajectory obtained by unrolling the predicted controls through the system dynamics starting from the initial state $x_0$, i.e.,

$$\hat{x}_{t+1} = f(\hat{x}_t, \hat{u}_t), \quad \text{with } \hat{x}_0 = x_0, \tag{2}$$

and let $\tau_x^\star = \{x_t^\star\}_{t=1}^{H}$ be the target state trajectory.

We define the control loss as

$$\mathcal{L}_{\text{control}} = \frac{1}{H} \sum_{t=0}^{H-1} \|\hat{u}_t - u_t^\star\|^2, \tag{3}$$

and the state loss as

$$\mathcal{L}_{\text{state}} = \frac{1}{H} \sum_{t=1}^{H} \|\hat{x}_t - x_t^\star\|^2. \tag{4}$$

Our total loss function combines these two components:

$$\mathcal{L} = \mathcal{L}_{\text{control}} + \lambda \, \mathcal{L}_{\text{state}}, \tag{5}$$

where $\lambda$ is a weighting factor that balances the contributions of the control loss and the state loss.

By incorporating the state loss $\mathcal{L}_{\text{state}}$, we encourage the predicted control trajectory to produce a state trajectory that remains close to the target state trajectory, thereby mitigating compounding errors during rollout.

As we show in Table 6, incorporating state trajectory loss helps mitigate these types of error accumulation and improve long-horizon trajectory accuracy. More specifically, we see that (1) as the prediction horizon increases, from $H = 9$ (Cart-pole) to $H = 40$ (Driving), so does the difference between using and not using state loss, (2) The gap between One-Off and Sequential also increases thus we do not generalize as well, (3) For the single-output regression model, the difference is even greater.

Table 6: Mean Cost Comparison for One-Off and Sequential Optimization (With and Without State Loss)

| | | One-Off Optimization | | | | Sequential Optimization | | | |
| | | Cart-pole | | Driving | | Cart-pole | | Driving | |
| Method | $K$ | No SL | With SL | No SL | With SL | No SL | With SL | No SL | With SL |
|---|---|---|---|---|---|---|---|---|---|
| **Single Opt.** | | | | | | | | | |
| Regression | 1 | 11.88 ±0.79 | 11.19 ±0.80 | 440.97 ±74.92 | 74.23 ±7.69 | 11.93 ±0.29 | **6.18 ±0.47** | 590.50 ±89.06 | **70.62 ±7.38** |
| MISO WTA | 32 | 10.62 ±0.77 | 10.48 ±0.77 | 128.32 ±22.55 | **30.17 ±2.24** | 1.86 ±0.20 | **0.83 ±0.06** | 151.4 ±20.70 | **30.75 ±2.15** |
| MISO mix | 32 | 10.63 ±0.77 | 10.48 ±0.77 | 153.37 ±21.8 | **33.95 ±2.39** | 2.28 ±0.26 | **0.79 ±0.04** | 212.70 ±28.02 | **33.38 ±2.21** |
| **Multiple Opt.** | | | | | | | | | |
| MISO WTA | 32 | 10.25 ±0.75 | 10.29 ±0.76 | 144.51 ±19.51 | **30.87 ±2.30** | 0.97 ±0.06 | **0.76 ±0.05** | 158.71 ±20.94 | **30.48 ±2.07** |
| MISO mix | 32 | 10.24 ±0.75 | 10.29 ±0.76 | 196.09 ±31.43 | **33.52 ±2.35** | 1.14 ±0.12 | **0.63 ±0.02** | 214.56 ±28.02 | **34.85 ±2.64** |

Table 6 shows a comparison between using and not using state loss (SL) across One-Off and Sequential Optimization settings. While both strategies benefit from including the state loss, the improvement is more profound in the Sequential Optimization setting. This is because it involves executing each solution and starting the next optimization from the resulting state, causing the compounding errors to accumulate. The One-Off Optimization setting also benefits from state loss, but the impact is less apparent and thus was not marked in the table.

### A.8 MODE FREQUENCY

Figure 7 presents a heatmap illustrating the percentage of selections for each specific mode (output) in MISO winner-takes-all by the selection function, which in this context chooses the output with the lowest resulting cost. Although a few modes appear to dominate, all other modes remain active, i.e., with a frequency greater than zero, indicating that there are problem instances where these less frequent modes identify the optimal action, thereby contributing to the overall performance improvements. Additionally, Fig. 8 compares the distribution of selections between MISO winner-takes-all and MISO mix with the expected, approximately uniform distribution of the Ensemble method.

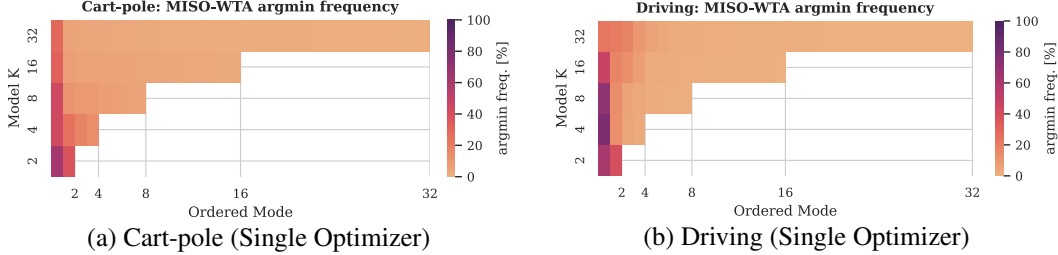

(a) Cart-pole (Single Optimizer)    (b) Driving (Single Optimizer)

Figure 7: Heatmap of MISO winner-takes-all outputs argmin frequency

### A.9 MEAN COST SEQUENTIAL OPTIMIZATION

Figure 9 shows the mean cost of each method with varying values of $K$, for both cart-pole and reacher environments. Both showcase that our method scales effectively and consistently with the number of predicted initial solutions and outperforms other approaches across varying values of $K$.

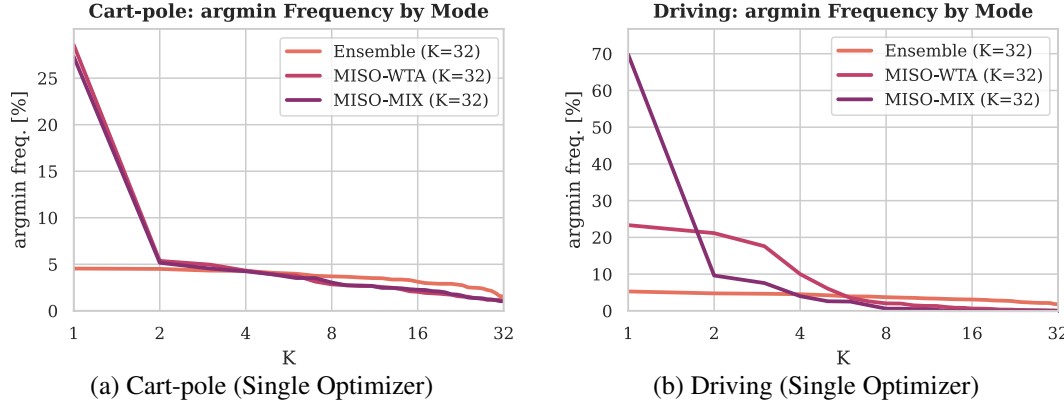

Figure 8: Argmin frequency of various methods for $K = 32$

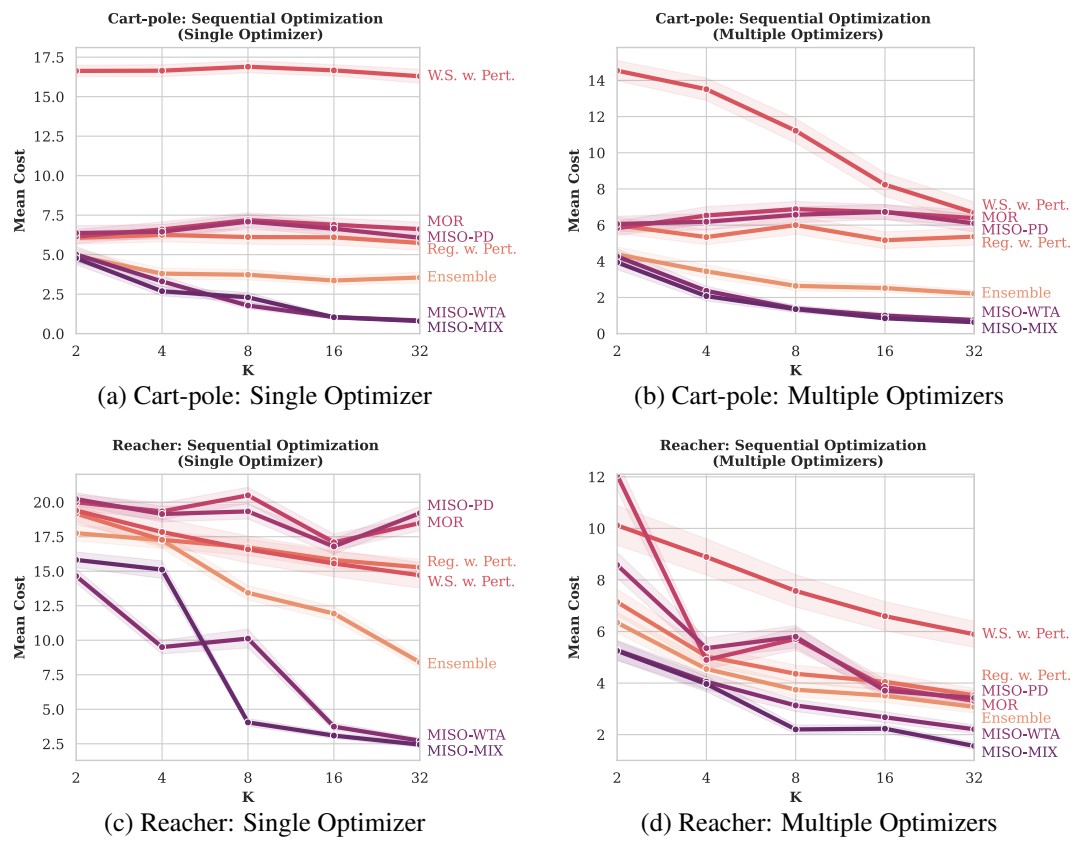

Figure 9: Mean Cost of the cart-pole and reacher environments with varying values of $K$. Subfigures (a) and (b) show the results for cart-pole using Single and Multiple Optimizers, respectively, while (c) and (d) display the same for the reacher environment. The shaded regions around each curve represent the standard error of the mean.

## A.10 SELECTION FUNCTION $\Lambda$

In our framework, the selection function $\Lambda$ plays a critical role in choosing the most promising initial solution from the set of candidates predicted by our model. While using the objective function $J$ as $\Lambda$ is a natural and effective choice, alternative choices for $\Lambda$ can be advantageous in certain scenarios.

ALTERNATIVE SELECTION CRITERIA

**Constraint Satisfaction:** In some applications, especially those that are safety-critical, it is essential to ensure that certain constraints are satisfied by the initial solution, even if it means accepting a higher value of $J$. In such cases, $\Lambda$ can be designed to prioritize solutions that satisfy these constraints. For example:

$$\Lambda(\hat{\mathbf{x}}_k^{\text{init}}, \psi) = J(\hat{\mathbf{x}}_k^{\text{init}}, \psi) + \beta\, C(\hat{\mathbf{x}}_k^{\text{init}}, \psi), \tag{6}$$

where $C(\hat{\mathbf{x}}_k^{\text{init}}, \psi)$ measures the degree of constraint violation, and $\beta$ is a weighting factor that penalizes constraint violations.

**Robustness Measures:** $\Lambda$ can incorporate robustness criteria, selecting initial solutions less sensitive to model uncertainties or external disturbances. For instance, it could favor solutions that maintain performance across various scenarios.

**Contextual Adaptation:** The selection function can adapt based on the problem instance $\psi$. For example, in varying environmental conditions, $\Lambda$ could prioritize more conservative or aggressive solutions depending on the context or operational requirements.

LEARNING THE SELECTION FUNCTION

Instead of hand-crafting $\Lambda$, it can be learned from data. One approach is to model $\Lambda$ as a parameterized function, such as a neural network, and train it jointly with the predictor model or separately. The learning objective could be to maximize the overall performance of the optimizer when initialized with the selected solutions, potentially incorporating criteria like safety, robustness, or energy efficiency.

EXAMPLES

**Safety-Critical Control:** In autonomous driving, safety constraints such as maintaining a safe distance from obstacles are crucial. $\Lambda$ can prioritize trajectories that ensure safety over those that simply minimize time or fuel consumption. For example, it can assign infinite cost to any solution violating safety constraints, effectively excluding unsafe options.

**Adaptive Behavior:** In robotics, $\Lambda$ can select initial solutions that favor energy efficiency when the robot's battery is low or prioritize speed when tasks are time-sensitive. By incorporating the robot's current state or mission objectives into $\Lambda$, the system can adapt its behavior accordingly.

### A.11 SEQUENTIAL VS. CLOSED-LOOP AND ONE-OFF VS. OPEN-LOOP

In control settings, the terms *sequential* and *closed-loop* evaluations are often used interchangeably. However, in the context of general optimization problems, the notion of "closed-loop" may not always be applicable, as there may be no dynamic environment or feedback mechanism involved. Therefore, we adopt more general terminology—referring to *sequential* evaluations—to encompass scenarios where decisions are made in a sequence but without necessarily involving feedback from an environment.

On the other hand, the distinction between *one-off* and *open-loop* evaluations is subtle yet significant. In a *one-off* evaluation, we assess the optimizer's performance on individual problem instances without any interaction with an environment. This means the optimizer solves a static problem, and we can directly compare different methods on the same set of instances. In contrast, open-loop control involves sequentially executing actions in an environment without feedback.

### A.12 EVALUATION MODES: ONE-OFF AND SEQUENTIAL

In our experiments, we assess the performance of the methods using two evaluation modes:

**One-off Evaluation**: In the one-off evaluation, problem instances are uniformly sampled from the evaluation dataset (disjoint from the training dataset). Each method is tested on the same set of independently sampled instances, ensuring a fair comparison across methods. The optimizer solves each problem instance independently, without any interaction with the environment or influence

from previous solutions. This evaluation mode focuses on the optimizer's ability to find high-quality solutions for individual problems in isolation.

**Sequential Evaluation**: In sequential evaluation, the optimizer interacts with the environment across a series of time steps. Starting from an initial state sampled from the evaluation dataset (disjoint from the training dataset), at each time step the optimizer adjusts its decisions based on the evolving state. This mode evaluates the optimizer's performance in a dynamic, real-time setting, highlighting its ability to manage evolving states and adapt over time.

