# OpenReview forum: "Learning Multiple Initial Solutions to Optimization Problems"
_ICLR.cc/2025/Conference — Submitted to ICLR 2025_

### Official Review · Reviewer_M1FQ · 2024-10-29

**Soundness:** 2
**Presentation:** 3
**Contribution:** 2
**Rating:** 6
**Confidence:** 5

**Summary:**

The paper introduces the Learning Multiple Initial Solutions framework, a neural network-based method for generating diverse initial solutions to improve local optimizer performance in time-sensitive tasks, such as robotics and autonomous driving. MISO supports both single-optimizer and multiple-optimizers configurations, allowing flexibility in selecting or parallelizing initial solutions.

**Strengths:**

The paper introduces a framework that leverages a neural network to generate initial conditions for general optimization problems. This framework enhances optimization performance by selecting an initial condition that is generated by a neural network trained to encourage diversity while remaining close to the global optimum.

**Weaknesses:**

While the empirical results indicate that the proposed framework may enhance optimization performance, the overall concept could be considered somewhat straightforward. A more substantial theoretical analysis could add depth to the work, as there is no clear indication that the neural network-generated initial states will consistently yield improved results. Consequently, the framework's contribution may seem limited in its novelty and theoretical rigor.

**Questions:**

For line 223, why is the pairwise distance loss defined as a function of $x^*$? To encourage diversity, shouldn’t the pairwise loss be maximized instead?
Could you elaborate what does problem instance $\psi$ stand for?

---

> ### Author Response · Authors · 2024-11-15
>
> **Concern 1:** *"While the empirical results indicate that the proposed framework may enhance optimization performance, the overall concept could be considered somewhat straightforward."*
>
>
> Thank you for your feedback. While our framework may seem straightforward, it uniquely fills a significant and novel gap in optimization initialization strategies. The area of learning to warm-start optimizers is indeed active [1]. However, existing approaches typically focus on a single initialization. To our knowledge, our work is the first to employ a single network to generate multiple diverse initializations for local optimizers. This advancement is particularly important for time-constrained non-convex optimization problems where multiple local minima are present.
>
> [1] Brandon Amos. Tutorial on amortized optimization.
>
> ------
>
> **Concern 2:**  *"A more substantial theoretical analysis could add depth to the work, as there is no clear indication that the neural network-generated initial states will consistently yield improved results. Consequently, the framework's contribution may seem limited in its novelty and theoretical rigor."*
>
> Without guarantees on the generator, it is difficult to make solid theoretical claims; conversely, with such guarantees, the problem would be significantly simplified and perhaps trivial. Most methods in this research field are heuristic-based, aiming to improve performance rather than provide formal guarantees. Nevertheless, as noted in lines 202-205, our approach allows for the inclusion of traditional initialization methods, such as warm-start heuristics, as part of the set of initializations, e.g., as the $(K+1)$-th initialization. This means our method does not risk degrading performance compared to established practices, effectively providing a safety net. Our contribution demonstrates that learning multiple diverse initializations can empirically enhance optimizer performance across various tasks and optimizers. We have provided extensive empirical evidence to support this.
>
> ------
>
> **Question 1:**  *"For line 223, why is the pairwise distance loss defined as a function of ( \mathbf{x}^\star )? To encourage diversity, shouldn’t the pairwise loss be maximized instead? "*
>
> Thank you for catching this error and for your insightful question. You are correct—the pairwise distance loss should be defined between the predicted initial solutions $ \hat{\mathbf{x}}_k^{\text{init}} $, and we aim to maximize this pairwise distance to promote diversity among them. The inclusion of $\mathbf{x}^\star$ in the loss function was a typographical error in the paper.
> In our experiments, we indeed considered the negative pairwise distances, effectively maximizing the distances between the predicted initial solutions. This encourages the model to output initializations that are different from each other, enhancing the exploration of the solution space by the optimizer. The error in the paper is solely a typo, and our implementation correctly reflects the intended formulation.
>
> ------
>
> **Question 2:** *"Could you elaborate what does problem instance stand for?"*
>
> Yes, we have detailed this in lines 291-293 of the paper:
>
> *"The problem instance parameters $\psi$ encompass the initial state $s_0$, and other domain-specific variables that parameterize the objective function or constraints, such as target states, reference trajectories, obstacle positions, friction coefficients, temperature, etc."*
>
> **We appreciate your careful consideration of our work and hope that our responses have adequately addressed your feedback.**

---

> > ### Comment · Reviewer_M1FQ · 2024-11-19
> >
> > Thank you for clarifying the confusions and addressing the weaknesses, I will adjust the rating.

---

### Official Review · Reviewer_1RPU · 2024-11-01

**Soundness:** 4
**Presentation:** 3
**Contribution:** 2
**Rating:** 6
**Confidence:** 5

**Summary:**

The paper considers the problem of improving optimization solvers. The paper proposes to learn multiple initial solutions for the optimizers to improve its final performance. The authors argue the learning of initial solutions should consider both the distance to ground-truth optimal parameters and the dispersion among the multiple initial solutions to ensure coverage of the optimization landscape. Experiments show the proposed method improves the optimization performance with three different optimizers on three simulated tasks. Although the paper is sound and well-written, I find the method missing important details, its novelty questionable and the experiment domains relatively simple.

**Strengths:**

1. The paper is very well written and easy to follow.

2. The figures, especially Figure 1 and 2, help clearly illustrate the proposed approach vs. prior methods.

3. The problem is well motivated and could be general for many tasks.

**Weaknesses:**

1. Important details about the method is missing: Section 4 focuses on introducing the learning objective, i.e., the loss function, However, it is unclear to me what function approximator was used to output the multiple initial guesses for the optimization problems. Is it a neural network with randomness that will have a different output in each forward pass? Is it a neural network outputting all K guesses in one forward pass and the output is split into K guesses? Or other design? This is arguably the most important part of the method but is missing.

1.1. The $\Lambda$ function is another important part of the method, but was very briefly described. According to Line 191 (“A reasonable choice”), the $\Lambda$ function is just chosen as the argmin of objective function? It is unclear what exactly is the choice of $\Lambda$ in the experiments.

1.2. The distance function in Line 223 only makes sense if the norm of the different dimensions of the parameter x make sense. For example, if one dimension of x ranges from [0, 1] and another dimension ranges from [0, 10^5], the norm is not a good measure of difference. It is unclear whether this assumption is satisfied in the experimented problems.

2. The claims in Line 259-262 may not be true. It depends on how the ensemble of models are learned. If the models themselves are multimodal such as latent models or energy-based models, they could represent multimodal behavior.

3. Missing information in the experiments:

3.1. What are the variations \psi for the experiment domains? This is completely missing and how do the authors make sure the training setup and evaluation setup do not overlap? Otherwise, the initial solution learning is just memorizing the best solutions for the tasks.

3.2. How much data is used to train in the experiments?

3.3. I believe the two evaluation modes in Line 359-367 are just open-loop and closed-loop. Should the authors use these well-accepted naming conventions instead?

3.4. Because Cartpole and Reacher are relatively simple domains, it is surprising to see just naïve optimizers cannot solve these problems very well. Did the authors constrain the optimization steps to certain budget such that optimal solution was not found in time?

4. One arXiv paper has done a similar approach on a more challenging racing task [1]. I understand it is an arXiv paper recently and the authors might have missed it, but it seems the paper has even accomplished what this paper proposes in future work (reinforcement learning tuning for the initial guess). Could the authors compare and state how the proposed method is still unique?

[1] Li, Z., Chen, L., Paleja, R., Nageshrao, S., & Gombolay, M. (2024). Faster Model Predictive Control via Self-Supervised Initialization Learning. arXiv preprint arXiv:2408.03394.

**Questions:**

See weakness

---

> ### Author Response · Authors · 2024-11-15
>
> **Concern 1:** *What function approximator was used to output the multiple initial guesses?*
>
> As described in lines 174-175, our method employs a neural network that outputs all $K$ initializations in a single forward pass, aligning with the mathematical definition in line 177.
>
> ------
>
> **Concern 2:** *What is the exact choice of $\Lambda$ used in the experiments?*
>
> In lines 190-191, we note that in our experiments, we use the objective function J as $\Lambda$, which is both natural and effective, as it directly aligns with our optimization goals. Furthermore, lines 192-194 explain how $\Lambda$ can be tailored to include additional criteria based on domain-specific requirements or further context from the problem instance $\psi$. In addition, we have added a section in the appendix (lines 1004-1042) that elaborates on the selection of $\Lambda$.
>
> ------
>
> **Concern 3:** *Is the distance function in Line 223 appropriate given possible differences in the scales of the dimensions of $x$, and is this assumption satisfied in the experimental problems?*
>
> Before training, all features are standardized (760-761).
>
> ------
>
> **Concern 4:** *Do the claims in Lines 259-262 consider latent models or energy-based models?*
>
> At the beginning of the example (lines 247-249), we confine the discussion to regression models or ensembles of regressors. We aimed to illustrate why explicitly promoting multimodality in multi-output regression models is essential.
>
> ------
>
> **Concern 5:**  *What are the variations $\psi$ in the experimental domains, how much data is used for training, and how do you ensure that the training and evaluation setups do not overlap?*
>
> 1. Variations ( $\psi$ ): We provide a detailed description in lines 782-784.
> 2. Training Data Size: Approximately 500,000 training instances were used for each task. We've added this detail in line 758.
> 3. Training and Evaluation Separation: As highlighted in line 50, the evaluation set contains previously unseen problem instances; specifically, the evaluation set is generated from scenarios different from the test set. To further clarify that, we've included a description of the evaluation settings in the appendix (1058-1071).
>
> ------
>
> **Concern 6:** *I believe the two evaluation modes are just open-loop and closed-loop*
>
>
> We appreciate your suggestion. Indeed, in control settings, closed-loop and sequential evaluations are identical. However, the term closed-loop is not always applicable in general optimization, which is why we use a more general terminology. One-off evaluation, however, is not the same as open-loop control. We've added a section in the appendix (1044-1056) to clarify this subtle difference.
>
> ------
>
> **Concern 7:** *Did the authors constrain the optimization steps or budgets such that naïve optimizers could not find the optimal solution in time for the Cart-pole and Reacher tasks?*
>
> Yes, exactly. While Cart-pole and Reacher are classic benchmarks, optimally solving all instances under strict runtime constraints can become non-trivial [1]. To reflect these realistic conditions, we imposed limits on the optimizers in our experiments (detailed in lines 681-784).
> These constraints make the problems more difficult, emphasizing the importance of good initializations to find high-quality solutions quickly. Our settings are not arbitrarily strict but reflect practical scenarios, such as autonomous driving, where strict runtime constraints are essential.
> All baseline methods were evaluated under the same constraints to ensure fair comparison.
>
> [1] Scokaert, et al. Suboptimal model predictive control.
>
> ------
>
> **Concern 8:** *"One arXiv paper [1] has done a similar approach on a more challenging racing task. Could the authors compare and state how the proposed method is still unique?"*
>
> Thank you for pointing out this related work. However, the paper [1] does not present novel concepts beyond existing research already cited (111-137).
> Key distinctions between our approach and [1] include:
>
> 1. Diverse Initializations: Unlike [1], which focuses on a single initialization, our method is designed to generate multiple diverse initializations. To the best of our knowledge, our work is the first to use a single network to learn multiple diverse initializations for local optimizers.
> 2. Broader Evaluation: The referenced paper evaluates their approach on a single racing task with a specific optimizer. In contrast, we demonstrate the effectiveness of our method across multiple tasks and various optimizers.
> 3. Innovative Methodology: Our approach introduces specialized loss functions to promote multimodality without relying on reinforcement learning.
> 4. Future Work: We propose integrating reinforcement learning to further enhance our method by predicting multiple initializations (472-473).
>
> **Thank you for your detailed comments. We hope that our responses have satisfactorily addressed your concerns.**

---

> > ### Comment · Reviewer_1RPU · 2024-11-24
> >
> > Thank you very much to the authors for providing direct answers to my concerns and questions in a short time. I am satisfied about most clarification the authors provided.
> >
> > Just one follow-up about concern 1: yes, I get it is neural network, but it is very important to know what architecture this neural network is - like the authors mentioned, "our method is designed to generate multiple diverse initializations. To the best of our knowledge, our work is the first to use a single network to learn multiple diverse initializations for local optimizers." Sharing what neural network architecture enables outputting diverse initialization is the key of the paper. Would you agree? I actually found this answer in Supplementary A.3 Network Architecture. I just thought the main paper could discuss the design of architecture for diffferent domains, which will be very helpful for practitioners.
> >
> > Based on the author's provided clarifications, I have raised my score to be 5 (marginally below the acceptance threshold).

---

> ### Author Response · Authors · 2024-11-24
>
> We appreciate your positive feedback and are glad our clarifications addressed your concerns.
>
> Regarding your follow-up:
>
> - *"[...] It is very important to know what architecture this neural network is [...] Sharing what neural network architecture enables outputting diverse initialization is the key of the paper. Would you agree? I actually found this answer in Supplementary A.3 Network Architecture."*
> - *"I just thought the main paper could discuss the design of architecture for diffferent domains, which will be very helpful for practitioners."*
>
> We emphasize that the neural network architecture itself does **not** enable diverse predictions. The novelty of our work lies in the **proxy training objectives** and the **selection function** (lines 53, 80, 96, 210–211), which, we believe, are independent of specific architectures and thus broadly applicable.
>
> To address your feedback, we've updated the revision to mention that architecture details are in the appendix (lines 174–175).
>
> **If there are remaining concerns preventing a higher score, please let us know, and we'd be happy to address them.**

---

> > ### Comment · Reviewer_1RPU · 2024-11-24
> >
> > Thank you to the authors for quick replies! Yes, I understand the architecture itself is not part of the novelty, but it is an important information to know which kinds of neural network enables learning diverse predictions (i.e., expressive enough to do so). I appreciate the authors taking this into consideration and revise the manuscript. I have revised the score to be 6: marginally above the acceptance threshold. Thank you for your effort during the discussion period!

---

### Official Review · Reviewer_LSK9 · 2024-11-03

**Soundness:** 3
**Presentation:** 3
**Contribution:** 2
**Rating:** 5
**Confidence:** 3

**Summary:**

The authors present an approach for computing multiple initializations for a downstream optimization task. The authors employ a winner-takes-all and pairwise distance loss to ensure multimodality of the initializations. This way, the produced initializations achieve better coverage and are more representative of a landscape with multiple optima.

**Strengths:**

The paper is easy to read and well-motivated. The presented approach is relevant, interesting and seemingly novel.

**Weaknesses:**

The proposed approach does not have any theoretical guarantees.

It is unclear if the experimental section considers varying constraints and environment parameters outside the autonomous driving example. The paper would also benefit from comparing to more well-known control examples, e.g., from Mujoco.

The choice of \Lambda beyond the optimization problem is somewhat unclear and lacks motivation.

The second optimization pipeline is merely an extension of the first, though it is presented as one of the major contributions.

**Questions:**

At first sight, the second approach presented by the authors seems to always be at least as good as the first one. The only limitation seems to be that it is more computationally expensive because multiple optimizers are used.

Why would any function other than J be used to specify \Lambda? It feels to me like J is simply poorly chosen if \Lambda is specified using any other criterion.

The name for the winner-takes-all is somewhat confusing. Intuitively, the name indicates that the best solution is used to compute the loss, not the worst.

In lines 299/300, what do the authors mean by "training loss over either control, state, or state-control sequences"? Does this mean that they use a distance metric for L_reg that takes the state into account? I would be helpful to have an explicit mathematical formulation in the appendix.

Is the difference between one-off and sequential evaluation the same as open-loop and closed-loop control? How are the initial conditions selected for the one-off evaluation?

As I understood it, the environment parameters do not vary between evaluations in the reacher and cart-pole settings. It would interesting to see how the approach performs, e.g., with varying constraints, obstacles and environment parameters (weight, arm length, etc.)

---

> ### Author Response · Authors · 2024-11-15
>
> **Concern 1:** *The proposed approach lacks theoretical guarantees.*
>
> Without guarantees on the generator, it is difficult to make solid theoretical claims; conversely, with such guarantees, the problem would be significantly simplified and perhaps trivial. Most methods in this research field are heuristic-based, aiming to improve performance rather than provide formal guarantees. Nevertheless, as noted in lines 202-205, our approach allows for the inclusion of traditional initialization methods, such as warm-start heuristics, as part of the set of initializations, e.g., as the $(K+1)$-th initialization. This means our method does not risk degrading performance compared to established practices, effectively providing a safety net. Our contribution demonstrates that learning multiple diverse initializations can empirically enhance optimizer performance across various tasks and optimizers. We have provided extensive empirical evidence to support this.
>
> ------
>
> **Concern 2:** *Unclear if constraints vary in experiments*
>
> As you correctly noted in Question 6, the environment parameters in the Reacher and Cart-pole tasks do not vary. We provided a comprehensive breakdown of each task (674-728) and referenced it within the experiment section (315-316).
>
> ------
>
> **Concern 3:** *Comparing to more examples*
>
> As detailed in our problem setup (144–154) and core concept (165–167), our research is task-agnostic. As such, rather than focusing on its nominal performance, we primarily concentrate on improving an optimizer's performance compared to that nominal performance. Using different tasks will primarily evaluate the optimizer's robustness and adaptability, which is not the focus of our work.
>
> ------
>
> **Concern 4:** *The choice of $\Lambda$ unclear and lacks motivation.*
>
> Thank you for your feedback. We have added a section in the appendix (1021-1049) that discusses choices for $\Lambda$, provides examples of how it can be designed to incorporate additional criteria such as safety constraints or robustness measures, and explains how $\Lambda$ can be learned or adapted based on the problem context.
>
> ------
>
> **Concern 5:** *"The second optimization pipeline is merely an extension of the first but is presented as a major contribution."*
>
> We appreciate your perspective but respectfully disagree. Unlike prior methods that predict only a single initialization, our approach seamlessly supports both single-optimizer and multiple-optimizers settings. This flexibility enables the utilization of multiple initial solutions in diverse ways, greatly enhancing the applicability and effectiveness of our method.
>
> ------
>
> **Question 1:** *The multiple-optimizers setting is always at least as good as the single-optimizer*
>
> Yes, our results show (415-416) that it does leads to consistently better results. However, our objective was not to compare these frameworks but rather to introduce two strategies for leveraging multiple predicted initial solutions. Parallelism in optimization is an active research area (128-140), yet most works still rely on a single-optimizer framework. Importantly, our results demonstrate that even when using a single optimizer, selecting an initialization from $K$ predictions significantly outperforms methods that rely on a single initialization.
>
> ------
>
> **Question 2:** "winner-takes-all naming"
>
> As detailed in lines 227-228, the Winner-Takes-All (WTA) method indeed selects the best prediction (the one with the lowest loss) and computes the loss using only that prediction. This is further defined mathematically in line 229, ensuring that the naming accurately reflects its functionality.
>
> ------
>
> **Question 3:** *Training loss over state sequences*
>
> To further clarify this point, we have elaborated the state loss section in the appendix (890-942) and added an explicit mathematical formulation of the loss functions in that state loss.
>
> ------
>
> **Question 4:** *sequential vs. closed-loop and one-off vs. open-loop*
>
> Thank you for this question. Indeed, in control settings, closed-loop and sequential evaluations are identical. However, the term closed-loop is not always applicable in general optimization, which is why we use a more general terminology. One-off evaluation, however, is not the same as open-loop control. We've added a section in the appendix (1044-1056) to clarify this subtle difference.
>
> ------
>
> **Question 5:** *Initial conditions in the one-off evaluation*
>
> We've included a description of the one-off setting evaluation in the appendix (1058-1071).
>
> ------
>
> **Question 6:** *Varying constraints and environment parameters in experiments*
>
> Similar to Concern 3. Introducing different constraints and obstacles would primarily evaluate the optimizer's robustness and adaptability, which is different from the focus of our current goals.
>
> **Thank you for your insightful review. We hope that our responses and revisions have adequately addressed your feedback.**

---

> > ### Author Response · Authors · 2024-11-26
> >
> > Dear LSK9,
> >
> > Thank you once again for taking the time to review our paper and for your insightful feedback!
> > As the author-reviewer discussion period is coming to an end, we hope that our clarifications have addressed your concerns and strengthened the manuscript.
> > If you feel your concerns have been resolved, we would greatly appreciate it if you could reflect this in your evaluation.
> > Should you have any remaining questions or need further clarification, please let us know—we are more than happy to answer.
> >
> > Best Regards,
> > The Authors

---

> > > ### Author Response · Authors · 2024-12-01
> > >
> > > Dear Reviewer LSK9,
> > >
> > > As we approach the end of the discussion period, we wanted to follow up regarding our paper. We have made every effort to address the concerns and questions you raised and have revised the manuscript accordingly.
> > >
> > > **Please let us know if you have any further questions or need additional information.**
> > >
> > > Best regards,
> > > The Authors

---

### Official Review · Reviewer_dukR · 2024-11-03

**Soundness:** 2
**Presentation:** 3
**Contribution:** 2
**Rating:** 6
**Confidence:** 3

**Summary:**

The paper introduces a method for learning a set of candidate initial solutions to warm-start optimal control problems. It proposes a series of objectives that encourage learning multimodal solutions, using a transformer architecture as a backbone to predict K control trajectories. The proposed method is evaluated on 3 different sequential optimization tasks and yields performance improvements over the presented baselines.

**Strengths:**

- The problem setting is relevant for the community. The proposed method outperforms the presented baselines on 3 different tasks and the trained neural network architecture achieves fast inference rates, which are suitable for closed-loop control.

- The paper is well-written overall and presents multiple ablations for the different loss functions that were introduced. The experiments also indicate that the trained model captures the multimodality of the solution space, to some degree, depending on the hyperparameter K that denotes the fixed number of initial solutions that are predicted.

**Weaknesses:**

- Using ensembles of models is not a very strong baseline for multimodality. Diffusion Policies [1] or action chunking transformers [2] might be stronger baselines. Even if they do not have such a fast inference time as the proposed method, it would further strengthen the paper to position the method with respect to such baselines.
- The method is only evaluated on 3 low-dimensional problems and it is unclear how its performance will scale or degrade in more complex settings.

[1] Cheng Chi et al, Diffusion Policy: Visuomotor Policy Learning via Action Diffusion.
[2] Tony Zhao et al. Learning Fine-Grained Bimanual Manipulation with Low-Cost Hardware.

**Questions:**

- As mentioned by the authors, finding optimal solutions for higher dimensional problems might be challenging. In that setting, the optimizers are more likely to provide only suboptimal solutions within a time budget.  How could the proposed loss be extended to leverage the potentially suboptimal generated by the optimizers in a more complex setting?

- Is there any specific reason why a specific optimizer wash was chosen for each task? The paper mentions a trivial generalization to a setting with a heterogeneous set of optimization methods. But no experiments are reported. Would a heterogeneous set of optimization methods increase the chances of finding multimodal solutions and thus help with scalability to more complex settings?

- What is the total time required to get a  "best solution"? (Fig 1). The inference time of the network might be fast but if the predicted initial solutions are not close to a local optimum, the optimizers might require multiple iterations to converge. Reporting the average number of additional iterations of the optimizers or the time to converge after warm-starting can potentially further highlight the quality of the learned initial solutions.

---

> ### Author Response · Authors · 2024-11-15
>
> **Concern 1:**
> *(1) Diffusion Policies or action chunking transformers. (2) Ensembles of models are not very strong baseline.*
>
> While these are attractive alternatives, achieving fast inference times is due to strict runtime constraints (see lines 11, 29, 153, and 284). Our method efficiently generates multiple initial solutions, requiring little to no additional inference time. We chose ensembles as a baseline because they utilize the same architecture, allowing for a fair comparison of our proposed loss functions and selection strategies.
>
> ------
> **Concern 2:**
> *The method is only evaluated on 3 low-dimensional problems*
>
> As detailed in our problem setup (lines 144–154) and core concept (lines 165–167), our research is task-agnostic. As such, rather than focusing on its nominal performance, we primarily concentrate on improving an optimizer's performance compared to that nominal performance. Introducing different constraints and obstacles would primarily evaluate the optimizer's robustness and adaptability, which is not the focus of our work. In addition, as detailed in lines 287-288, since our method interacts with the optimizer rather than directly with the control task, it handles sequences of states and control inputs ($H \times S$, $H \times A$), which moves beyond simple state-action pairs.
>
> ------
> **Question 1:**
> *How can the proposed loss be extended to leverage suboptimal optimizer solutions in higher-dimensional problems?*
>
> Thank you for this insightful question. In our current setting, we do not use optimal solutions but rather (near)-optimal solutions obtained from local optimizers with extended runtimes (lines 754-758). Nevertheless, if only suboptimal solutions are available, we can adapt our loss function to account for solution quality, e.g., by incorporating a weighting factor based on the suboptimality gap—if quantifiable—we can prioritize learning from higher-quality solutions while still leveraging all available data.
>
> ------
> **Question 2:**
> *Why was a particular optimizer chosen for each task?*
>
> Thank you for raising this point. We selected DDP for the cart-pole task based on [1], where this optimizer was introduced. MPPI was chosen for reacher-hard due to its additional control dimension [2], and iLQR as the standard optimizer for the nuPlan autonomous driving benchmark.
>
> [1] Brandon Amos, et al. Differentiable MPC for end-to-end planning and control.
> [2] Yuval Tassa, et al. Deepmind control suite.
>
> ------
> **Question 3:**
> *A heterogeneous set of optimization methods / why were no experiments conducted on this?*
>
> As we mention on line 481, exploring a heterogeneous set of optimization methods is an exciting direction for future work. Incorporating different optimizers could be beneficial, as each optimizer has strengths and weaknesses and may perform differently depending on the problem characteristics. Using a heterogeneous set allows us to capitalize on these differences, potentially increasing the chances of finding diverse, high-quality solutions.
>
> ------
> **Question 4:**
> *"What is the total time required to get a 'best solution'? (Fig 1)"*
>
> In our experiments, the total time to achieve the best solution includes the neural network inference and the optimizer runtime, demonstrating that our approach is fast enough for real-time applications (lines 723-728). If you were specifically referring to $\Lambda$, the optimizers, by default, assign a scalar cost to each solution, making the selection of the minimum cost solution computationally negligible.
>
> ------
> **Question 5:**
> *Reporting the average number of additional iterations of the optimizers or the time to converge can potentially further highlight the quality of the learned initial solutions*
>
> While examining optimizer convergence could offer insights, metrics such as average additional iterations or convergence time are not particularly meaningful in our context. For instance, an optimizer might converge more quickly within the allocated budget but reach a suboptimal local minimum. In contrast, another optimizer might use the entire budget to find a better solution. Faster convergence does not necessarily equate to better performance. Additionally, there are no practical advantages to converging before the budget limit, as the optimizer would wait the remaining time until that solution is required. Furthermore, sample-based optimizers like MPPI operate with a fixed sample budget, eliminating the concept of iterations.
>
> Our main emphasis is on the quality of solutions within the given computational constraints, which is directly reflected in the cost metrics we report. We demonstrate that our method scales efficiently and consistently with the number of predicted initial solutions $K$ (lines 855-888), and we find that all outputs remain effective even as $K$ increases (lines 946-953).
>
> **We thank you for your constructive feedback and hope our revisions meet your expectations and alleviate any remaining concerns.**

---

> > ### Author Response · Authors · 2024-11-26
> >
> > Dear dukR,
> >
> > Thank you once again for taking the time to review our paper and for your insightful feedback!
> > As the author-reviewer discussion period is coming to an end, we hope that our clarifications have addressed your concerns and strengthened the manuscript.
> > If you feel your concerns have been resolved, we would greatly appreciate it if you could reflect this in your evaluation.
> > Should you have any remaining questions or need further clarification, please let us know—we are more than happy to answer.
> >
> > Best Regards,
> > The Authors

---

> > > ### Comment · Reviewer_dukR · 2024-11-28
> > > **Score update**
> > >
> > > I’ve read the author’s response and appreciate their effort in addressing my questions and concerns. I have adjusted my score accordingly to 6.
> > >
> > > As multimodality is a central feature of the approach, I encourage the authors to extend their related work and better position MISO wrt other approaches that have shown the ability to capture multimodality even if they are not as fast.
> > >
> > > Furthermore, my concerns about the limited evaluation stand and prevent me from raising the score even higher. Additional experiments on more complex settings might shed light on the approach's scalability limits or even highlight its true potential. Either way, they would improve the quality of the paper.
> > >
> > > Given the recent success of leveraging MPPI for quadruped locomotion [1] or predictive sampling for Real-time Behaviour Synthesis[2], the proposed method might actually be able to scale to more complex settings. However, without further experiments, it's difficult to know if it can be used for 3D quadcopters,  6Dof manipulators, quadrupeds, or even humanoids.
> > >
> > > - [1] Giulio Turrisi et al. On the Benefits of GPU Sample-Based Stochastic Predictive Controllers for Legged Locomotion
> > > - [2] Taylor Howell, Predictive Sampling: Real-time Behaviour Synthesis with MuJoCo

---

> > > > ### Author Response · Authors · 2024-11-28
> > > >
> > > > Thank you for acknowledging our efforts to address your questions and concerns.
> > > >
> > > > 1. "*I encourage the authors to extend their related work and better position MISO wrt other approaches that have shown the ability to capture multimodality even if they are not as fast.*"
> > > >
> > > > We appreciate your suggestion, and while we cannot make significant revisions at this stage due to conference guidelines, we are committed to incorporating this in the final version.
> > > >
> > > > 2. "*[...] my concerns about the limited evaluation stand and prevent me from raising the score even higher. Additional experiments on more complex settings might shed light on the approach's scalability limits or even highlight its true potential. Either way, they would improve the quality of the paper.*"
> > > >
> > > > We acknowledge that additional experiments would further demonstrate the scalability and potential of our approach. Indeed, this is true for many research works, and ours is no exception, but we cannot introduce new experiments at this stage of the review process due to the guidelines.
> > > >
> > > > 3. "*[...] the proposed method might actually be able to scale to more complex settings. However, without further experiments, it's difficult to know [...]*"
> > > >
> > > > We share your excitement about the potential applications of MISO to other complex systems, and we are actively exploring these avenues as part of our future work. We believe that MISO's design, which is agnostic to the specific task, positions it well for scaling to more domains.
> > > >
> > > > That said, we believe that the experiments presented in our paper are adequate to evaluate the proposed approach and showcase its potential impact. In particular, the autonomous driving task using the nuPlan benchmark represents a realistic and complex problem with direct practical applications. This task involves complex trajectory spaces and dynamic interactions with other agents and requires real-time performance—characteristics that make it both challenging and representative of complex environments.
> > > >
> > > > We hope that our clarifications have addressed your concerns to some extent. If you feel that our explanations alleviate some of the limitations you've noted, we kindly ask you to consider this in your final evaluation.
> > > >
> > > > Thank you once again for your constructive feedback and for contributing to improving our work.

---

### Author Response · Authors · 2024-11-15

We thank all the reviewers for their thoughtful feedback and for recognizing the relevance of our problem setting, our experimental results' strength, and our paper's quality. We have carefully considered all the comments and believe that the concerns raised were minor and have been adequately addressed in our responses and revisions.

We hope our clarifications have resolved any uncertainties and strengthened the paper. If you feel that your concerns have been addressed, we would appreciate it if you could reflect this in your score or respond if there are any remaining concerns.

---

### Public Comment · ~Arun_Kumar_Singh1 · 2025-02-05
**A closely related work**

Hi

We have been working towards generating multiple solutions to an optimization problem with applications to swarm trajectory generation.

This is a pre-print of our work that leverages variational models with trajectory optimization to produce diverse solutions. Hope you find our work useful and if so, please cite us in your future works.

Swarm-Gen: Fast Generation of Diverse Feasible Swarm Behaviors
https://arxiv.org/html/2501.19042v1.

---

> ### Public Comment · ~Elad_Sharony1 · 2025-02-05
>
> Dear Arun Kumar Singh,
>
> Thank you for sharing your preprint and for your interest in our research.
>
> Since our work has been publicly available (https://arxiv.org/abs/2411.02158) since November 4, 2024—approximately three months prior to your January 31, 2025 preprint—we would appreciate it if you would cite our paper to acknowledge its contributions.
>
> We will likewise include a citation to your preprint in forthcoming revisions.

---

> > ### Public Comment · ~Arun_Kumar_Singh1 · 2025-02-07
> >
> > Absolutely, I am planning to combine your idea with ours as well.

---

> > ### Public Comment · ~Arun_Kumar_Singh1 · 2025-02-07
> > **Some additional prior work**
> >
> > We had a paper in IROS 2024, that could also generate multiple feaisble solutions in Autonomous Driving setting. The pre-print appeared in March 2024.
> >
> > https://arxiv.org/abs/2403.19461

---

> > > ### Public Comment · ~Elad_Sharony1 · 2025-02-09
> > >
> > > Thank you for sharing your IROS 2024 paper.
> > >
> > > While we appreciate learning about new methods for generating feasible trajectories, MISO has some distinct elements: It employs a fully amortized warm-starting model to predict multiple and diverse initial solutions *simultaneously* (rather than sampling). This allows the parallel execution of multiple optimizers while relying on off-the-shelf optimizers without learning the optimization process itself. These design choices arise from applications with strict runtime constraints, where rapidly generating multiple promising initializations is essential.
> > >
> > > We look forward to seeing how these complementary approaches might synergize in future work!

---

### Public Comment · ~Anjian_Li1 · 2025-02-12
**A related work**

Dear authors,

We have been exploring a similar approach for accelerating non-convex optimization using amortization. In our recent paper DiffuSolve, we used a diffusion model to sample from the distribution of local optima, which were then used as diverse initial guesses to warm-start the solver in parallel. Given the similarities between our topics and approaches, we would appreciate it if you could acknowledge our work in your paper. Thank you.

DiffuSolve: Diffusion-based Solver for Non-convex Trajectory Optimization
[https://arxiv.org/abs/2403.05571](https://arxiv.org/abs/2403.05571)

---

> ### Public Comment · ~Elad_Sharony1 · 2025-02-13
>
> Dear Anjian Li,
>
> Thank you for sharing your work on DiffuSolve. While both of our studies aim to accelerate non‐convex trajectory optimization by providing high-quality initial guesses, there are notable differences in our approaches. DiffuSolve leverages a diffusion model that iteratively samples candidate solutions, whereas our method directly predicts *multiple* diverse initializations in a **single forward pass**. This design is particularly well-suited for applications with strict runtime constraints, where rapidly generating multiple promising initializations is critical.
>
> We believe these complementary perspectives open exciting avenues for future research, including the potential for hybrid approaches that combine the strengths of both methods. Given the relevance of our respective contributions, we will include appropriate citations to your work in our next revision.

---

> > ### Public Comment · ~Anjian_Li1 · 2025-02-15
> >
> > Dear Elad,
> >
> > Thank you for your response. Your approach, which focuses on real-time applications using transformers, presents a valuable perspective, while our work emphasizes sampling from the solution distribution with diffusion models. This is an exciting area and hybrid methods that combine the strengths of both approaches could be interesting to explore for future research.

---

### Meta-Review · Area_Chair_N1Tb · 2024-12-26

**Metareview:**

This paper presented a novel framework for predicting multiple initial solutions for sequential black-box optimization problems.
The authors introduced two strategies to predict initial solutions called single-optimizer and multiple-optimizers (not necessarily running in parallel), for the chosen of the best one-step optimizer.
As a neural network driving approach, the proposed method does not have any theoretical analysis, which is ok. The authors evaluated three distinct sequential optimization tasks in simulation, without comparison to any decision-transformer line of algorithms.
The improvement of either theoretical or empirical performance are welcome.

**Additional Comments On Reviewer Discussion:**

After discussion, the reviewers' provided neutral feedback.

---

### Decision · Program_Chairs · 2025-01-22

Reject